# Thetan Berserker: Fast And Stochastic Distance-based Clustering

## Abstract

Clustering is a challenging NP-hard problem. Polynomial approximations are of paramount importance for identifying intriguing hidden representations of data at reasonable execution times. In this work, we propose a novel clustering algorithm called Thetan Berserker (TB). TB is a centroid-based clustering method controlled by a single distance parameter. TB revitalizes an old family of sequential algorithms, which are adored for their speed but are known to be order-sensitive. In addition, TB enables widely used algorithms such as KMeans and DBSCAN by improving their initial conditions. Theoretical aspects are provided in detail, along with extensive comparisons and benchmarks. Examples of real world applications are provided using publicly available data of different dimensionalities. A wide range of performance boosts in clustering accuracy, memory usage, and runtime are reported. By dramatically reducing clustering ambiguities while staying at incredibly low complexity, TB creates a new standard for clustering.

## 1 Introduction

Clustering has been the epitome of AI research for more than half a century because, when achieved, it can infer the underlying structure of the data without any annotations, substantially improving automation. It has a range of applications across the fields of science and medicine. For example, collaborative filtering (Ungar & Foster, 1998), trend analysis (Aghabozorgi et al., 2015), LLMs (Tirumala et al., 2023), computer vision (Caron et al., 2018), social networks (Mishra et al., 2007), biological data analysis (Zhao & Karypis, 2005) and signal processing (Orhan et al., 2011). In addition, inference based on clustering is of cardinal importance as it is often applied as the first step in ML pipelines. Information retrieved from clustering is fed into subsequent learning algorithms in many applications such as recommendation systems (Lu et al., 2015), medical analytics (Xu & Wunsch, 2010), and detection of unexpected patterns (Agrawal & Agrawal, 2015).

At the same time, clustering still stands as an extremely hard algorithmic problem (NP-hard). Its main challenges include: a) dealing with clustering ambiguity, e.g. clusters mixing; b) tackling order sensitivity i.e. incorrect outputs that depend on data ordering; c) estimating the correct number of clusters; d) long execution times, e) major memory needs, f) large number of hyper-parameters and f) handling of outliers.

Our primary interest in proposing a new method stems from the fact that, in nature, we often have some prior knowledge of the clusters we intend to find, such as the physical dimensions of atoms, cells, animals, etc. In such cases, it is much more useful to infer the number of clusters rather than setting a fixed limit on the number of clusters in the data. For this reason, we propose a new approach, Thetan Berseker (TB), which uses a single hyperparameter. Nonetheless, TB outperforms the state-of-the-art in accuracy, speed, and robustness in more than 30 experiments across dimensions and domains.

## 2 Related Work

Given the wide range of applications, the problem of clustering has been tackled using different approaches such as dimensionality reduction, density estimation, probabilistic methods, spectral methods, and distance-based techniques. Among all these methods, distance-based techniques have been used extensively. Distance-based methods can be separated in four categories. Those that cluster with assumptions concerning a) the maximum number of clusters ( KMeans (Lloyd, 1982),

KMeans++ (Arthur & Vassilvitskii, 2007)), Bisecting KMeans (Steinbach & Karypis, 2000; Di & Gou, 2018), b) a distance threshold (Hierarchical Clustering (HC) (Murtagh & Legendre, 2014)), c) cluster density (MeanShift (Comaniciu & Meer, 2002), DBSCAN (Ester et al., 1996; Schubert et al., 2017)) or d) any combination of the above (HDBSCAN (Campello et al., 2013; McInnes & Healy, 2017)). Another separation can be due to the type of problems that they can try to solve: a) Linearly-separable (KMeans), b) Nonlinearly-separable (DBSCAN), or c) both (Hierarchical Clustering). A third divide can be due to the way they process the data: a) sequentially (process each sample as they arrive) or b) offline (process the entire dataset).

Researchers have worked to figure out distance-based clustering for linearly separable problems for more than 60 years. The idea that we may be able to approximate clustering solutions started getting attention in the 1950s (due to KMeans) and HC in the 1960s. A few decades after sequential algorithms such the Leader Algorithm (Rush & Russell, 1988), BIRCH (Zhang et al., 1996), BSAS (Theodoridis & Koutroumbas, 2006), MBSAS, TTSAS (Real et al., 2014) and SL (Patra et al., 2011) appeared. In addition, some of these methods, such as QuickBundles (Garyfallidis et al., 2012), were successful in specialized domains but not used widely. The reason is that the results of these methods depend heavily on the order of sampling, and therefore results can change drastically from one run to the other. This is known as the ordering problem or order sensitivity. This is not an issue with only sequential algorithms. Many others, including KMeans, suffer from this problem.

Thetan Berserker (TB) is introduced here to dramatically reduce this problem while sustaining low complexity at the expense of a single parameter $\theta$. In contrast, BSAS has two parameters, KMeans, and MeanShift has three, etc.

In this paper, the focus will be primarily on TB tackling linearly separable problems, but TBSCAN will also be introduced, which can tackle nonlinear problems. TB will be challenged and compared across 30 experiments and more than 20 methods.

## 3 THETAN BERSERKER

The name Thetan Berserker (TB) is derived from the way the algorithm works with some inspiration from history. Thetan stems from the algorithm's single distance threshold $\theta$. Berserkers, in the context of this algorithm, are cluster modes that compete for spatial territory in data space.

TB's foundation stone is Thetan Sequential (TS), a straightforward sequential clustering approach. See Alg. 1. TS is in simple terms a simplified version of a basic sequential scheme. In short, TS will visit each data point only once and if a distance metric between a sample and the centroid is less than a threshold $\theta$ will enter the same cluster otherwise it will create a new cluster.

TS has multiple advantages: a) single pass - examining each feature only once, b) low time complexity, c) use of a single hyper-parameter, d) minimal memory footprint in contrast, for example, to Hierarchical Clustering, e) online execution - ideal for asynchronous or streaming systems and f) easy implementation. However, a major disadvantage of TS is the lack of stability when the order of selection changes in datasets with underlying dense clusters (manifestation of the ordering problem).

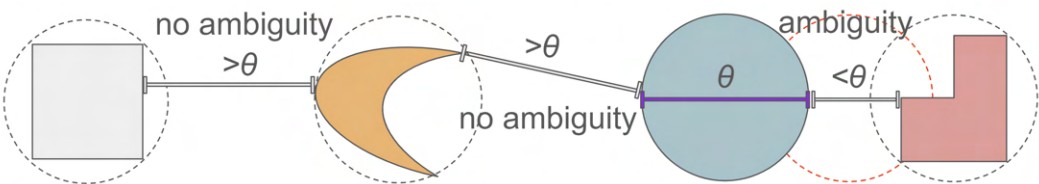

Figure 1: Geometric intuition. Assuming that each shape is filled with 2D points and an $L^2$ distance threshold matches the diameter of the 3rd shape. Alg. 1 will never mix the first two shapes due to an ordering issue. But it will mix points from the last two shapes. Alg. 2 will be able to separate also the last two without being affected by the ordering problem.

We propose a solution to this problem by studying the space around hypothetical clusters in our data (see Fig. 1). Note that if the closest distance between two points belonging to two different clusters is greater than $\theta$, a single run of TS is sufficient for any order due to the clusters being simply said, far enough from each other. But if that is not the case, then order sensitivity becomes important.

**Definitions**. We denote the number of samples $N \in \mathbb{Z}^+$, number of dimensions $D \in \mathbb{Z}^+$. The data sets are denoted with $X$ and contain feature vectors $x \in \mathbb{R}^D$. The Thetan threshold is denoted with $\theta \in \mathbb{R}_{\geq 0}$. The number of clusters is denoted with $K \in \mathbb{Z}^+$. A clustering result is denoted with $C$ and is represented with centroids $\mu$ and labels $\lambda$. $C_\xi$ denotes different clustering numbers (not individual clusters). In a addition a clustering $C$ contains clusters $c_1, c_2, \ldots, c_K$. For example, $C_2$ means second clustering. But cluster $c_3$ of $C_2$ has a single centroid $\mu_3$. The description of Thetan Sequential follows (see Alg. 1).

---

**Algorithm 1** Thetan Sequential (TS)

---

**Input:** Data $X$ of size $N \times D$ with samples $x_i$, $i \in [0, N-1]$ and hyper-parameter $\theta$
**Output:** Clustering $C$ of cardinality $K$ with centroids $\mu_k$ and labels $\Lambda$
$x_0 \in c_0, K \leftarrow 1$         ▷ First feature starts first cluster
  **for** $i = 1$ **to** $N - 1$ **do**
     distance_buffer $\leftarrow$ infinity$[K]$        ▷ Dynamic buffer holds distances from centroids
     **for** $k = 0$ **to** $K - 1$ **do**
        d $\leftarrow$ distance$(x_i, \mu_k)$        ▷ This is were metric evaluation takes place
        **if** $d <= \theta$ **then**
          distance_buffer$[k] \leftarrow$ d
     **end**
     $m \leftarrow$ min(distance_buffer)        ▷ Only the smallest distance is used
     $a \leftarrow$ argmin(distance_buffer)
     **if** $m <= \theta$ **then**
        $x_i \in c_a$        ▷ Assign to closest cluster and update centroid
     **else**
        $K \leftarrow K + 1$        ▷ Number of clusters grows
        $x_i \in c_{K-1}$        ▷ Create a new cluster
     **end**
**end**

---

In Thetan Berserker (see Alg. 2), the centroids of TS become an input to a second TS operating directly on these initial centroids. Then this second round of modes (after a low complexity clean-up - relabel function) is brought back to start a second iteration pre-pending the actual data $X$. This is a key point of the innovation. Therefore data which was initially $X$ becomes stack$(M, X)$ where $M$ contains previous centroids $\mu_k$, $k \in [0, K-1]$. TB converges very fast, and for this purpose, we use only a fixed and small number of iterations (max of 2 is used everywhere in this work).

---

**Algorithm 2** Thetan Berserker (TB)

---

**Input:** Data $X$ of size $N \times D$, hyper-parameter $\theta$, *ITER* = 2 fixed
**Output:** Clustering $C$ of cardinality $K$ with centroids $\mu_k$ saved in $M$ of size $K \times D$
counter $\leftarrow 0$
  **repeat**
     shuffle$(X)$        ▷ Randomly sample from the data
     **if** *counter = 0* **then**
        $C \leftarrow$ TS$(X, \theta)$        ▷ TS runs for first time here
     **else**
        $X \leftarrow$ stack$(M, X)$        ▷ Previous run centroids are placed in the beginning of $X$
        $C_2 \leftarrow$ TS$(X, \theta, $ metric$)$
        $C \leftarrow C_2$
        $X \leftarrow$ destack$(X, M)$        ▷ Remove extra Berserker centroids
     **end**
     $C_3 \leftarrow$ TS $(M, \theta)$        ▷ Cluster only new centroids $M$
     $C \leftarrow$ relabel$(C, C_3)$        ▷ Directly update labels providing a new clustering
     counter $\leftarrow$ counter $+ 1$
**until** *counter = ITER*;

---

Note that exactly the same parameter $\theta$ is used across Alg. 1 & 2. The function stack simply prepends the Berserker centroids to data $X$. destack removes these centroids to return the original data $X$. These added and removed centroids are the Berserker centroids because they have survived through a heavily stochastic process (due to the shuffle of the data and the myriad distance pulls). relabel updates the labels $\Lambda$ of $C$ using $C_3$ results. The size of $\Lambda$ is $N$. $M$ of size size $K \times D$ refers to a

matrix containing all centroids $\boldsymbol{\mu}_k$ of clustering $\boldsymbol{C}$. The default number of iterations is fixed to 2. This decision is confirmed by the ablation and iterations study (see Fig. 3).

## 4 THEORETICAL ASPECTS

We postulate the following theorems assuming $L^2$ is our norm of choice. First, let's identify which datasets would be exactly satisfied (reach a unique and global solution) by Alg. 1.

**Theorem 1**. Given a clustering problem $\boldsymbol{C}$ with clusters $c_1, c_2, \ldots, c_K$. If there are no pairs of samples $\mathbf{x}_i, \mathbf{x}_j$ from $c_i$ to $c_j$, $i \neq j$ where $\mathbf{x}_i \in c_i$ and $\mathbf{x}_j \in c_j$, that have a distance $< \theta$, then Alg. 1 will never mix clusters with a threshold parameter $\theta$ at exactly one pass. The order of the selection of the samples will have no effect on the final outcome as long as all features are used.

**Proof**. In Alg. 1. all the distances are computed either inside a cluster or between clusters. All the inside cluster distances will be $< \theta$, and all the distances between clusters will be $> \theta$. Therefore, there are no cases where clusters are created between the actual clusters given that there are no pairs of samples $\mathbf{x}_i, \mathbf{x}_j$ from cluster $i$ to cluster $j$ that have a distance $< \theta$. ■

This brings us to an equivalence relationship between how the algorithm performs and how close the hypothetical boundaries of the underlying clusters are.

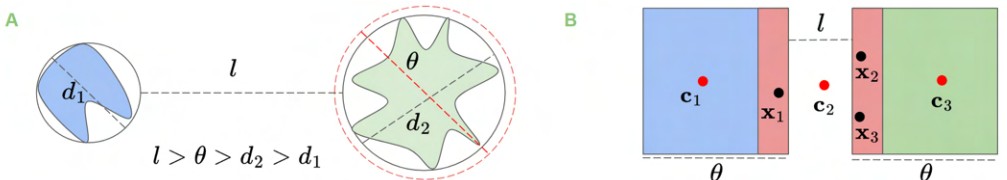

Figure 2: Visual guides. $l$ is the minimum distance between clusters. A) Under the condition above TS will always converge at the global solution in a single pass. Insensitive to order and unaffected by size or geometry (convex vs non-convex). B) Some orders are better than others. TB will resolve the correct clusters even in cases where TS has generated more clusters than necessary (3 shown with blue, red, and green colors rather than 2). Red dots represent TS centroids, black dots represent sampled points near the edges of two uniform distributions.

**Lemma 1** If there are no pairs of samples $\boldsymbol{x}_i, \boldsymbol{x}_j$ from cluster $c_i$ to $c_j$, $i \neq j$ where $\boldsymbol{x}_i \in c_i$ and $\boldsymbol{x}_j \in c_j$ that have a distance $l < \theta$, then due to Theorem 1 the order of selection will not affect the result. However, this also tells us that if the $L^2$ diameter of the circumscribed hypersphere is $< \theta$ then also the shape of the clusters will not affect the results. In other words, if the contours of the clusters are convex or non-convex, the outcome will be the same.

**Proof** Imagine two clusters $c_1$ and $c_2$ bounded (circumscribed) in hyperspheres of diameters $d_1$ and $d_2$. Assume that $d_2 > d_1$ (one hypersphere is larger than the other one). The clusters can contain any shape of points $\boldsymbol{x}$, convex or non-convex. The minimum distance between $c_1$ and $c_2$ is denoted with $l$. The optimal result would be if Alg. 1 could exactly find the two clusters. Anything more or less than two would be incorrect. Using proof by cases we can see that if $l < d_1, d_2$, and $\theta > d_1$, $\theta > d_2$ and $\theta > l$. Alg. 1 will not generate two clusters missing the global solution. The same would happen if $l > d_1, d_2$ and $\theta < l$. However, if $l > d_1, d_2$, and $\theta = l$ then Alg. 1 is guaranteed to find the two clusters. This is because all distances between points will be less than threshold $\theta$ only in the same clusters. Similarly, we would reach a global solution if $l > d_1, d_2$ and $\theta < l$ but $\theta > d_1, d_2$. In short, we now have the following condition where $l > \theta > d_2 > d_1$. In this condition, there is no point $\boldsymbol{x}$ that can be assigned to the wrong cluster. ■.

Although the proof is demonstrated for two clusters, the argument holds for an arbitrary number of clusters. There are specific sizes and interclass distances where the solution is exact, guaranteed, and single pass. Given that clustering is an NP-hard problem with very few theoretically backed ideas, we can agree that the fact that TS (Alg. 1) will always provide for these unique data sets for any order of selection is an important observation. Clearly, there are types of datasets that are ideal for TS, and some are not ideal. For example, those where the underlying clusters are close to each other. This is an area that Alg. 2 (TB) shines.

This is because the nonlinear optimization problem that we are usually trying to solve in centroid/distance-based clustering problems is expressed as $\min_{\mu_1, \ldots, \mu_k, z_{ij}} \sum_{i=1}^{k} \sum_{j=1}^{n} z_{ij} \|\boldsymbol{x}_j - $

$\boldsymbol{\mu}_i\|^2$ subject to $\sum_{i=1}^{k} z_{ij} = 1, \forall j \in \{1, \ldots, n\}$. $z_{ij} = 1$, if $\boldsymbol{x}_j$ is assigned to cluster $i$ or 0 otherwise. KMeans, for example, is an approximate (heuristic) solution. TS provides a different solution to this nonlinear problem by not using $K$ but a distance threshold $\theta$. Therefore, the assignments are not violated, and still $\sum_{i=1}^{k} z_{ij} = 1, \quad \forall j \in \{1, \ldots, n\}$. However, $K$ is inferred on the go, and it can only grow at increments of one, $K \in \{1, \ldots, \infty\}$. Similarly, there is an additional constraint that $i$ (index of data increases monotonically $i \in [1, \ldots, N]$ and stops at a single pass from the data.

**Lemma 2** The centroids of a dataset are a reduced representation of the original data. Representing data as their centroids increases empty space (a proxy for sparsity).

**Proof** Given that the centroids are local averages in an ideal scenario, they would exactly approximate the data with one centroid for each sample or one centroid per two or more samples. In that way, we always partition the space to be either equal or less than the data. In other words, the centroids are like an infinite shrinkage of the clusters to a point. Moving from one representation to another increases empty space between the clusters. ∎

See Fig. 2A for a visual guide. At this stage, it is important to understand that some orders of selection are better than others. Assume, for example, a grid of uniformly distributed clusters with circumscribed diameters $d$ at equal interclass distances $l$ where $l < d$. We could simply sort the coordinates $x, y$. That order would be preferred over a random order of selection because the first order would allow Alg. 1 to converge to the global solution in one pass. A random order could generate centroids appearing on empty space between the actual clusters simply because two boundary points could be picked first.

**Theorem 2** Starting with the centroids of Alg. 1 helps Alg. 2 reach an improved solution (less incorrectly assigned samples).

**Proof** We will show this for two uniform distributions at distance $l$ (see visual guide at Fig. 2B). The sides of each distribution are equal to $\theta$. Let's assume that $l < \theta$. A selection order where the first points selected were $\boldsymbol{x}_1, \boldsymbol{x}_2$ and $\boldsymbol{x}_3$ (close to the edges of the two clusters) is guaranteed to create unnecessary extra clusters. This is because Alg. 1 will create a new centroid at $\boldsymbol{\mu} = (\boldsymbol{x}_1 + \boldsymbol{x}_2 + \boldsymbol{x}_3)/3$. However, if we start with $\boldsymbol{c}_1, \boldsymbol{c}_2$ and $\boldsymbol{c}_3$ Alg. 1 is forced to create two enduring centroids $\boldsymbol{\mu}_1 = (\boldsymbol{c}_1 + \boldsymbol{c}_2)/2$ or $\boldsymbol{\mu}_2 = (\boldsymbol{c}_2 + \boldsymbol{c}_3)/2$. Note that the order of selection of centroids (123, 213, or 312) does not change the outcome. ∎

This generalizes for an arbitrary number of clusters. See details in sections A.10-A.12. Due to Theorems 1-2 and Lemmas 1-2, TB is adding a new constraint that TS cannot satisfy. The constraint is $\|\boldsymbol{\mu}_i - \boldsymbol{\mu}_i\| > \theta$. Centroids will be far from each other, providing better coverage and a reduced number of clusters.

**Complexity analysis**. Alg. 1 has a worst-case time complexity (upper bound) $\mathcal{O}(NKD)$ that depends on the number of samples N and the number of estimated clusters K and number of dimensions D. We assume here that most of the computation is from the calculation of distances between samples and centroids. The worst-case complexity takes place when every data point belongs to a different cluster (all singleton clusters). In such an event, the worst time complexity is $\mathcal{O}(DN^2)$. The best time complexity is $\mathcal{O}(ND)$ when only one cluster. Assuming most memory is spent on saving centroids and labels, Alg. 1 has a best case (lower bound) space complexity of $\mathcal{O}(N)$ and worst case (upper bound) of $\mathcal{O}(N)$. The highest bound is when all clusters are singleton clusters. However, in most cases $K << N$. Therefore, TS requires a remarkably small amount of memory. TB (Alg. 2) builds on top of TS (Alg. 1), but now the time complexity also depends on the number of iterations $I$. However, everywhere in this work, we fixed $I = 2$. Therefore, Alg. 2 is of the same complexity as Alg. 1. In addition, TB's cleanup operations work in the space of centroids or small label updates. Therefore, they do not change the order of complexity. In short, TB's time and space complexity is the same as that of TS. This is also experimentally shown in Tab. 1. Proofs are available at A.8 and A.9.

**Other algorithmic contributions**: I) In order to compare TB, we introduce a simple iterative version of TS. TSR (Thetan Sequential Randomized) is a version where we simply run TS multiple times (default 10) for different shuffles of the data, collect all their centroids, re-cluster them, and reassign the data to the last round of centroids. II) In order to increase our understanding of the differences between BSAS and KMeans, we introduced a new algorithm TBK (TB seeding + KMeans), which starts with TB and then gets the centroids of the K biggest clusters and provides them as initialization

Table 1: Comparisons between clustering algorithms. Highlight identifies top performers.

| | Method | AC ↑ | | NMI ↑ | | SIL ↑ | | FMS ↑ | | ARS ↑ | | Clusters | | Runtime ↓ | | Memory ↓ | |
|---|---|---|---|---|---|---|---|---|---|---|---|---|---|---|---|---|---|
| | | mean | std | mean | std | mean | std | mean | std | mean | std | mean | std | mean | std | mean | std |
| 1 | BIRCH | 0.571 | 0.057 | 0.956 | 0.004 | 0.509 | 0.008 | 0.872 | 0.015 | 0.871 | 0.015 | 296.0 | 5.8 | 7.518 | 0.103 | 5.400 | 0.516 |
| 2 | BSAS | 0.155 | 0.020 | 0.949 | 0.003 | 0.502 | 0.008 | 0.857 | 0.012 | 0.857 | 0.012 | 300.0 | 0.0 | 5.999 | 0.065 | 155.000 | 0.000 |
| 3 | CLARANS | 0.012 | 0.008 | 0.877 | 0.003 | 0.282 | 0.008 | 0.574 | 0.010 | 0.568 | 0.010 | 300.0 | 0.0 | 5726.562 | 1632.025 | 27.000 | 0.000 |
| 4 | CURE | 0.814 | 0.026 | 0.964 | 0.001 | 0.542 | 0.001 | 0.920 | 0.002 | 0.919 | 0.002 | 300.0 | 0.0 | 2518.961 | 375.720 | 174.000 | 0.000 |
| 5 | EM_GMM | 0.931 | 0.008 | 0.971 | 0.001 | 0.536 | 0.002 | 0.927 | 0.003 | 0.927 | 0.003 | 300.0 | 0.0 | 138.870 | 9.507 | 1375.000 | 0.000 |
| 6 | FCM | 0.131 | 0.024 | 0.922 | 0.002 | 0.403 | 0.004 | 0.703 | 0.005 | 0.698 | 0.006 | 300.0 | 0.0 | 1014.085 | 366.602 | 3091.000 | 0.000 |
| 7 | MBSAS | 0.152 | 0.019 | 0.949 | 0.003 | 0.502 | 0.008 | 0.858 | 0.012 | 0.857 | 0.012 | 300.0 | 0.0 | 6.692 | 0.120 | 155.000 | 0.000 |
| 8 | OPTICS | 0.786 | 0.066 | 0.300 | 0.030 | -0.628 | 0.027 | 0.067 | 0.001 | 0.003 | 0.000 | 63.3 | 5.8 | 739.002 | 94.169 | 188.000 | 0.000 |
| 9 | TB | 0.997 | 0.002 | 0.976 | 0.000 | 0.556 | 0.001 | 0.953 | 0.001 | 0.953 | 0.001 | 300.9 | 0.7 | 0.349 | 0.005 | 8.000 | 0.000 |
| 10 | TBK | 1.000 | 0.000 | 0.976 | 0.000 | 0.556 | 0.001 | 0.954 | 0.001 | 0.954 | 0.001 | 300.0 | 0.0 | 0.869 | 0.008 | 8.000 | 0.000 |
| 11 | TS | 0.862 | 0.023 | 0.971 | 0.001 | 0.530 | 0.005 | 0.938 | 0.004 | 0.937 | 0.004 | 320.7 | 4.3 | 0.161 | 0.002 | 1.000 | 0.000 |
| 12 | TSR | 0.994 | 0.005 | 0.976 | 0.001 | 0.555 | 0.001 | 0.953 | 0.001 | 0.953 | 0.001 | 301.5 | 1.4 | 7.829 | 0.056 | 4.000 | 0.000 |
| 13 | TTSAS | 0.145 | 0.027 | 0.949 | 0.004 | 0.482 | 0.014 | 0.861 | 0.014 | 0.861 | 0.014 | 318.7 | 5.6 | 6.444 | 0.127 | 156.000 | 0.000 |
| 14 | BISECTING | 0.218 | 0.014 | 0.931 | 0.000 | 0.436 | 0.002 | 0.778 | 0.001 | 0.776 | 0.001 | 300.0 | 0.0 | 0.787 | 0.030 | 13.000 | 0.000 |
| 15 | DBSCAN | 0.825 | 0.014 | 0.722 | 0.002 | 0.100 | 0.004 | 0.083 | 0.001 | 0.019 | 0.000 | 301.0 | 0.8 | 1.355 | 0.016 | 26.100 | 0.316 |
| 16 | HDBSCAN | 1.000 | 0.001 | 0.887 | 0.001 | 0.414 | 0.002 | 0.295 | 0.003 | 0.195 | 0.003 | 300.0 | 0.0 | 132.364 | 3.192 | 184.000 | 0.000 |
| 17 | KMEANS | 0.522 | 0.075 | 0.945 | 0.006 | 0.478 | 0.015 | 0.818 | 0.026 | 0.817 | 0.027 | 300.0 | 0.0 | 2.394 | 0.229 | 8.400 | 0.516 |
| 18 | KMEANS++ | 0.855 | 0.035 | 0.968 | 0.002 | 0.535 | 0.005 | 0.919 | 0.009 | 0.919 | 0.009 | 300.0 | 0.0 | 2.921 | 0.215 | 20.000 | 0.000 |
| 19 | KMEDIANS | 0.104 | 0.014 | 0.942 | 0.003 | 0.478 | 0.008 | 0.820 | 0.013 | 0.819 | 0.013 | 300.0 | 0.0 | 31.176 | 0.241 | 506.000 | 0.000 |
| 20 | MEANSHIFT | 0.999 | 0.002 | 0.976 | 0.001 | 0.556 | 0.001 | 0.954 | 0.002 | 0.953 | 0.002 | 299.9 | 0.3 | 1014.352 | 23.430 | 37.300 | 1.418 |
| 21 | MEANSHIFT++ | 0.033 | 0.009 | 0.720 | 0.000 | 0.312 | 0.003 | 0.291 | 0.000 | 0.166 | 0.000 | 33.1 | 0.3 | 3.802 | 0.059 | 8.000 | 0.000 |
| 22 | XMEANS | 0.591 | 0.037 | 0.951 | 0.003 | 0.494 | 0.007 | 0.846 | 0.012 | 0.845 | 0.013 | 300.0 | 0.0 | 76.938 | 4.836 | 158.000 | 0.000 |
| 23 | TBSCAN | 0.997 | 0.003 | 0.975 | 0.000 | 0.555 | 0.001 | 0.953 | 0.001 | 0.953 | 0.001 | 300.8 | 1.0 | 3.320 | 0.019 | 8.000 | 0.000 |

for KMeans. This is a task of improving KMeans seeding. III) Because TS and TB are primarily focused on linearly separable problems, we introduce an algorithm to deal with density-based nonlinear clustering problems. We call this algorithm TBSCAN. Basically, we start with TB, and then the output centroids become input to a modified DBSCAN version (with additional parameters epsilon and min_samples to control for nonlinear shapes). In summary, we introduce 1 major algorithm, TB, and 4 supporting ones (TS, TSR, TBK, and TBSCAN). Extensive comparisons follow.

## 5 RESULTS

All results were performed on a single thread of a single CPU. No GPUs were used in this work. We compare our methods with other well-known clustering methods. Various metrics are used for evaluation including Normalized Mutual Information (NMI), Silhouette score (SIL), Fowlkes-Mallows Score (FMS), and Adjusted Rand Score (ARS). Because clustering evaluation can often be ambiguous, we also introduce a stricter metric, Apparent Centroid distance (AC) (see details in A.1). All experiments, including the compared methods and the evaluation, were done using scikit-learn (Pedregosa et al., 2011), pyclustering (Novikov, 2019), meanshift++ (directly from author's GitHub) and skfuzzy packages (for fuzzy-cmeans). Tracemalloc module was used to measure peak memory. Runtime was reported with Python's time package. The Thetan methods were developed in C (via Cython). All methods have underlying C or C++ implementations via Pythonic interfaces. Due to the large number of experiments, most of the results and details are available in Appendix A.

### 5.1 SIMULATION EXPERIMENTS

We randomly sample from multi-variate normal distributions with mean 0 and identity covariance matrix. Each distribution contains 500 samples. The centers of the distributions are set on a $30 \times 10$ grid. Therefore, we have a total of 300 ground truth clusters. Each distribution center is at a distance of 5 units from its closest neighboring center. This creates a dense clustering setting which will be challenging for most algorithms. The total number of points (samples) in this experiment is $150,000$. **Ablation Study**. In Fig. 3A, we use the setup above to study if the different parts of the TB algorithm are actually improving overall accuracy. As Alg. 2 suggests, TB is split into 4 parts: I) TS, II) CL, III) TS2, and IV) CL2, where TS stands for Thetan Sequential and CL stands for cluster new centroids and update labels. The experiment was repeated 20 times. As we can see in the boxplots of Fig. 3, every part of the algorithm clearly improves overall accuracy. **Convergence Study**. In Fig. 3B, we examine the hypothesis saying that TB may need only two iterations. In this experiment, we check if repeating iterations can actually improve the results. As shown in the violin plot of Fig. 3B repeating TB iterations does not improve NMI scores. TB01 is the default version with 2 iterations. The experiment goes up to TB14, which has 15 iterations. The statistics shown are from 20 repetitions. In short, TB does convergence in only 2 iterations for the experiment at hand. **Inter-class distance trade-off**. In this experiment, we study how algorithms perform as the inter-cluster distances shrink or grow. See Fig. 3C. Here the minimum distances across two clusters change from 3 to 10. As expected, all methods have trouble when the distances between clusters are small, and that gradually

improves as the distances grow. The parameters used are TB ($\theta$ 3.6), HDBSCAN (min_samples 40), KMeans++ (K 300, tol. 0.0001), and MeanShift (bandwidth 2.6).

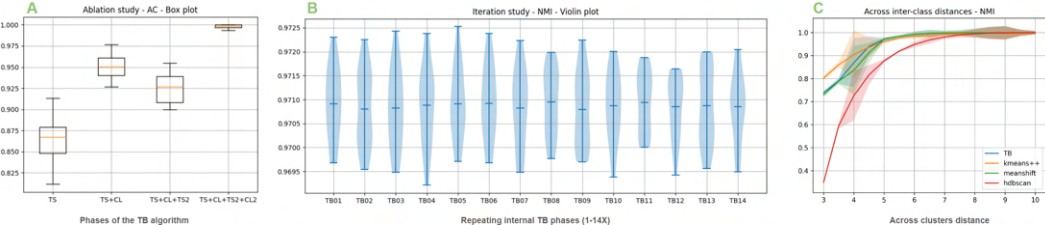

Figure 3: Study of trade-offs. A) Ablation study shows that all parts of TB contribute to its performance. B) Iteration study shows fast convergence. C) NMI improves as clusters become more distant from each other.

**Large comparisons**. We evaluated more than 20 methods in clustering the 300 clusters as described above. All parameters used are in section A.3. The statistics shown are after 10 repetitions for each method. Results are summarized on Tab. 1 and Fig. A4-A15. Thetan Berserker (TB) is one of the best-performing methods. TS is only 2X faster than TB, but it identifies the wrong number of clusters and has low evaluation scores. HDBSCAN generates comparable scores (AC) but is 200X slower. KMeans++ takes 4X more time but makes estimation mistakes. MeanShift takes a lot longer, 1000X more time. DBSCAN is only 2X slower than TB. However, it does not achieve high evaluation scores. Therefore, we consider TB to perform well for such datasets. In addition, TSR has higher scores than TS, but it generates a less accurate number of clusters than TB. TBK outperforms KMeans++ (stands for KMeans with KMeans++ seeding (Arthur & Vassilvitskii, 2007)). TBSCAN is faster than HDBSCAN by 48X. BSAS, MBSAS, TTSAS, and BIRCH lose accuracy across many evaluation metrics. In contrast, TB has a unique performance balancing memory, runtime, and accuracy. Visual plots are available in Fig. A4-A15. TB continued performing at the highest level in experiments with outliers and uniform distributions of equal or varying scales (see Fig. A31- A45).

**Predicting $\theta$**. TB has only one parameter, while most of the methods have at least two or three parameters. Nonetheless, two important questions emerge: a) Would it be possible to find $\theta$ automatically from the data? b) How fast? Finding $\theta$ often depends on what someone wants to do, and therefore, it cannot always be found automatically. However, apart from established techniques such as the Elbow method (Liu & Deng, 2021), researchers can use random walks effectively. In more detail, by storing the distances between consecutive points we can create a footprint of the dataset. This is shown in Fig. 4A. The distributions of the random walks can regress the distance across clusters. This is a method that is highly efficient ($\mathcal{O}(ND)$). Note that each distribution is clearly distinguishable from another. Here the minimum interclass distance is from 5 to 45 units. Other more advanced methods include the Auto Elbow Method (Onumanyi et al., 2022), Gap Statistic (Tibshirani et al., 2001) and maximizing the Silhouette Score (Rousseeuw, 1987), but they require a larger number of samples.

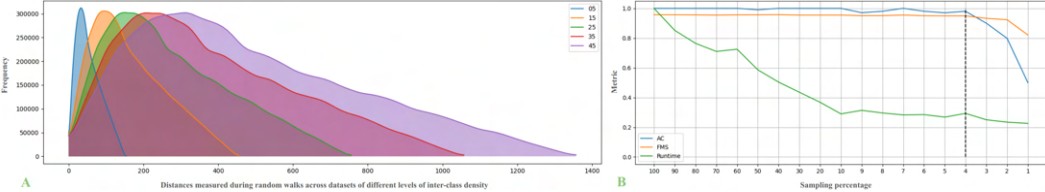

Figure 4: Predicting $\theta$ and sub-sampling. A) Linear-time random walks can be used to infer good estimates for hyper-parameter $\theta$. Here are the distributions for growing inter-class distances of 5, 15, 25, 35, and 45 units. A pattern emerges. Details at section A.13. B) TB seems robust in sampling parts of the data. Here we sample from 100% down to 1%. Accuracy stays high until after 4% while runtime is reduced. This is evidence of robustness. Runtime is normalized by dividing with an initial runtime of 83ms at 100%.

**Robustness in sub-sampling**. Another important way to measure the robustness of an algorithm is to see how stable it can be while reducing the actual data. In this experiment, we start with the same setup as in the previous section, but now we measure accuracy after keeping from 100% down to 1% of the data. See Fig. 4. Samples are removed randomly. Accuracy measured by AC metric stays high until after 4% while runtime is reduced because less data are being used. Runtime is normalized to fit

the plot by dividing with an initial runtime of 83ms. Holding AC after removing 96% (100-4) of the data is strong evidence of robustness to large density changes.

## 5.2 STANDARDIZED BENCHMARKS

The widely used and publicly available clustering benchmarks framework (Gagolewski, 2022) is applied here. Eight linearly and eight nonlinearly-separable datasets have been processed using TB and TBSCAN, respectively. As seen on Tab. A1, TB achieves high clustering performance on linearly separable datasets with minimal runtime, while TBSCAN shows overall high accuracy on non-linear datasets. Parameters used are reported on Tab. A4. For this evaluation, we used the 2D embedding of the digits dataset from scikit-learn (Pedregosa et al., 2011), which consists of 1,797 samples, each with 64 features. Tab. A3 presents a comparison of various clustering methods, with TB emerging as the most efficient in terms of both runtime and peak memory usage. TB achieves the lowest runtime (0.0008s) and requires no additional memory (<1 MB), making it highly resource-efficient compared to the hierarchical clustering (HC) methods. While HC-WARD outperforms TB slightly in terms of clustering accuracy metrics like RI (0.90499) and ARI (0.51293), TB maintains competitive performance with respectable values across NCA (0.52226), RI (0.84255), and ARI (0.37490). Given its efficiency, TB offers a significant advantage in scenarios where resource constraints are critical while still delivering comparable clustering quality. More information is available in section A.6.

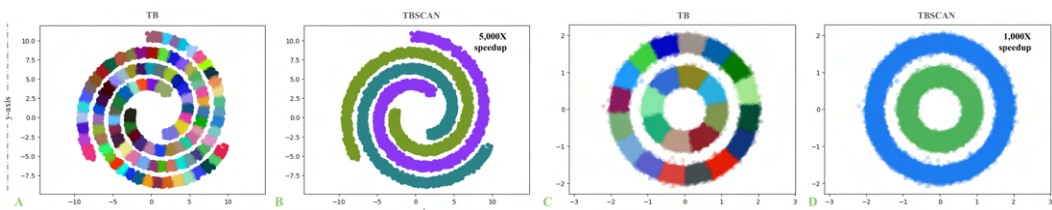

Figure 5: TB enables density-based approaches for nonlinearly-separable problems. Here shown with Spiral and Circles benchmarks. TB reads the x and y coordinates in a completely random fashion, but it is still able to evenly separate the clusters (A, C). TBSCAN is up to 5,000X faster than DBSCAN (B, D).

**Nonlinearly-separable benchmarks**. In this experiment, we used the benchmarks Spiral and Circles from Scikit-Learn. We increased the number of points of the datasets to go up to the level of hundreds of thousands. The purpose of this experiment is to examine the behavior of TB and TBSCAN on testbed nonlinear datasets. In Fig. 5, we see that TB is building equivariant parts, and then TBSCAN connects them to build the nonlinear parts. For the Spiral dataset, we used TB ($\theta$ 1.0) and TBSCAN (eps 0.5, min_samples 1). Number of points is 375,000. For the Circles dataset (1 million points), we used TB ($\theta$ 1.0) and TBSCAN (eps 0.8, min_samples 1). Additional details at A.6 and Fig. A24.

## 6 APPLICATIONS

### 6.1 SUPERPIXELS

Here we examine if TB can assist superpixel segmentation. NYUV2(Arbelaez et al., 2011) and BSDS500(Nathan Silberman & Fergus, 2012) datasets are used. Fig. 6 shows qualitative results. For TB, we assign the 2D spatial information and pixel intensity as features of each pixel for clustering. Felzenszwalbs' method (Felzenszwalb & Huttenlocher, 2004), SLIC(Achanta et al., 2012), Quickshift(Vedaldi & Soatto, 2008) and Compact watershed(Neubert & Protzel, 2014) were selected as a comparison method mainly for their availability. Despite the fact that our method is not tailored for superpixel purposes, it creates object boundary-compliant superpixels compared to other methods. All methods excluding TB were run using scikit-image(van der Walt et al., 2014), with the following parameters: 1. BSDS500 dataset: scale 300, sigma 1.0, minimum size 30 for Felzenszwalbs's method, number of segments 200, compactness 0.1, sigma 1 with automatic parameter estimation for SLIC, kernel size 7, maximum distance 30, ratio 0.5 for Quickshift and markers 125, compactness 0.00001 for Compact Watershed and $\theta$ 0.17 for TB. NYUV2 dataset: scale 75, sigma 1.0, minimum size 30 for Felzenszwalbs's method, number of segments 400, compactness 0.1, sigma 1 with automatic parameter estimation for SLIC, kernel size 5, maximum distance 30, ratio 0.5 for Quickshift and markers 200, compactness 0.00001 for Compact Watershed. For TB $\theta$ 0.19. Grid search with qualitative evaluation was used to select the optimal parameters. The input to TB was simply an array

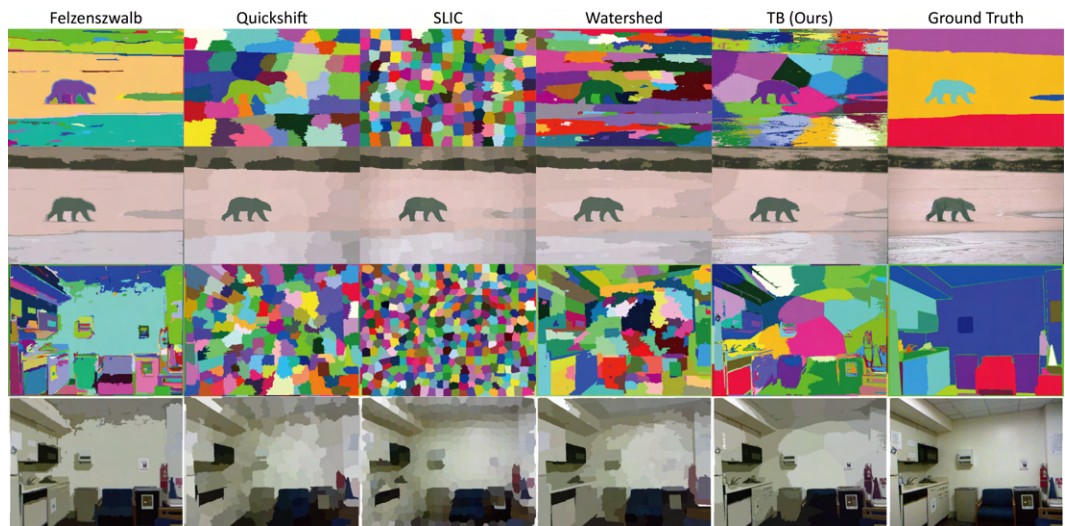

Figure 6: Example results from the BSD500 (top two rows) and the NYUV2 (bottom two rows) datasets.

$X$ holding the normalized position $(x, y)$ and LAB space intensity features (weighted to balance between spatial information) for each pixel in the image. Details are shown on Tab. 2. We calculate metrics suggested by Stutz et al. (2018). More specifically, the table contains scores for Boundary Recall (REC) (Martin et al., 2004), Normalized Undersegmentation Error and its variant (UE, UEB) (Van den Bergh et al., 2015; Neubert & Protzel, 2012), Explained Variation (EV) (Moore et al., 2008) and Compactness (CO) (Schick et al., 2012). Higher REC, EV, CO, and lower UE, UEB indicate more appropriate superpixels. Runtime and the number of superpixels are indicated in the table as well. As the BSDS500 dataset has multiple ground truth labels, the metric on each ground truth was averaged first before averaging the results of all images in the dataset. In both datasets, TB is capable of creating superpixels of reasonable quality and high average metrics. We want to emphasize that no further processing was done to modify the method to be superpixel friendly (e.g. smoothing kernels, connectivity regularizations). High metrics on the BSDS500, which had multiple ground truths, also suggest our method is capable of creating superpixels that are generalizable between different goals.

Table 2: Comparisons between TB and superpixel algorithms.

| Datasets | Algorithms | REC ↑ | | UEB ↓ | | UE ↓ | | EV ↑ | | CO ↑ | | Superpixels | | runtime (sec) ↓ | |
|---|---|---|---|---|---|---|---|---|---|---|---|---|---|---|---|
| | | mean | std | mean | std | mean | std | mean | std | mean | std | mean | std | mean | std |
| BSDS500 | Felzenszwalbs | 0.420 | 0.091 | 0.969 | 0.0135 | 0.225 | 0.340 | 0.0506 | 0.0324 | **2.44e-03** | 6.43e-03 | 112 | 47.3 | 0.147 | 5.75e-03 |
| | SLIC | 0.271 | 0.0592 | 0.980 | 8.06e-03 | **0.0112** | 0.0103 | 0.0648 | 0.0305 | 1.53e-05 | 1.06e-06 | 186 | 1.29 | 0.164 | 3.41e-03 |
| | Quickshift | 0.291 | 0.0903 | 0.979 | 0.0103 | 0.0728 | 0.0854 | 0.0569 | 0.0309 | 2.63e-04 | 1.26e-04 | 39 | 6.39 | 3.64 | 0.0782 |
| | Compact Watershed | 0.316 | 0.0829 | 0.977 | 0.0101 | 0.0565 | 0.0580 | 0.0566 | 0.0313 | 1.35e-04 | 4.01e-05 | 126 | 0.00 | **0.0736** | 4.55e-03 |
| | TB (Ours) | **0.745** | 0.0972 | **0.944** | 0.0222 | 0.0344 | 0.0326 | **0.0811** | 0.0300 | 1.27e-05 | 2.26e-05 | 87.6 | 21.0 | 0.388 | 0.0642 |
| NYUV2 | Felzenszwalbs | **0.540** | 0.0593 | **0.929** | 0.0175 | 1.41e-03 | 1.73e-03 | 0.0824 | 0.0159 | **5.18e-05** | 7.69e-05 | 580 | 178 | 0.334 | 0.0228 |
| | SLIC | 0.193 | 0.0212 | 0.975 | 6.69e-03 | 2.23e-04 | 1.56e-04 | 0.0712 | 0.0168 | 3.56e-06 | 1.31e-07 | 388 | 2.36 | 0.338 | 6.71e-03 |
| | Quickshift | 0.233 | 0.0349 | 0.970 | 7.93e-03 | **1.92e-04** | 1.76e-04 | 0.0756 | 0.0150 | 1.98e-05 | 4.67e-06 | 169 | 19.0 | 3.99 | 0.0647 |
| | Compact Watershed | 0.242 | 0.0396 | 0.968 | 8.49e-03 | 2.78e-04 | 2.15e-04 | 0.0736 | 0.0160 | 4.37e-05 | 9.47e-06 | 192 | 0.00 | **0.154** | 7.28e-03 |
| | TB (Ours) | 0.445 | 0.0800 | 0.941 | 0.0200 | 8.20e-04 | 6.60e-04 | **0.0872** | 0.0164 | 2.24e-05 | 2.07e-05 | 74.2 | 12.5 | 0.687 | 0.0942 |

## 6.2 PROCESSING 3D BRAINS

We examined if TB would be affected by the 3D structure of the brain. Here, we used T1 images from HCP (Van Essen et al., 2013) containing 1,200 subjects, which are widely common in MR experiments. These images can have millions of voxels and intricate underlying anatomy. HCP has multi-modal neuroimaging and behavioral data of young adult subjects aged 22-35 at 3 Tesla (3T). Dimensions are $260 \times 311 \times 260$ with a voxel resolution of 0.7 $mm^3$. The features used are the $x, y, z$ scanner coordinates of each voxel and intensity $w$ of the image. To balance the features we multiplied the $w$s by 3. Therefore, matrix $X$ is now a 2D array of shape 21,023,600 $\times 4$. $\theta$ value is 150. In Fig. 7, we see the results of this experiment. TB takes only, on average, 17.38 seconds to cluster this dataset. The clustering labels match well with the underlying anatomy (A-C). In addition, we built a new image from the labels where each voxel is replaced with its corresponding centroid $w$ value (B). The results match the original anatomy without blurring the edges. Notably, a single label

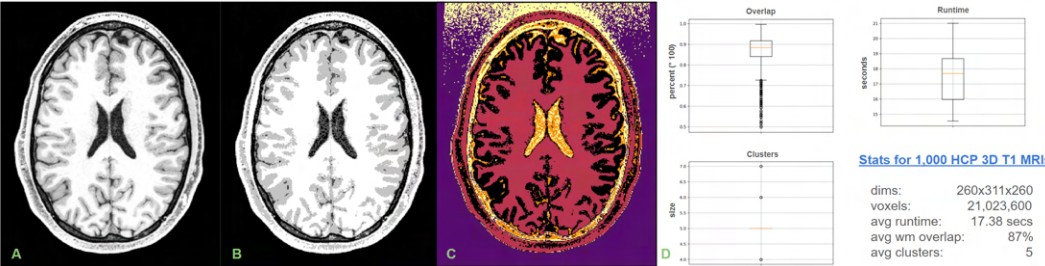

Figure 7: TB processing 1,000 3D brain images ($\theta$150). A) Original HCP data, subject 102008, b) Reconstructed image of the same from TB output (contains only 5 unique intensity values), c) Clusters found (5 clusters), D) Statistical analysis and summary for all participants. TB consistently produces 5 clusters keeping the white matter solid (87% overlap). Each image contains 21 million voxels and is clustered in 17.38 sec (avg).

is used for most of the white matter. The experiment shows that TB can process millions of features in seconds, providing meaningful results. The compression ratio achieved is 26.6X. 160MB (original size), 147 MB (GZip) and 6 MB (GZip after TB). More details, including results at different $\theta$ and comparisons against other methods are available at section A.18 (see Fig. A46).

### 6.3 PATTERN RECOGNITION IN TIME AND HIGH DIMENSIONS

A series of experiments studying the ability of TB to find repeating patterns in familiar signals and real publicly available ECG data (Wagner et al., 2020) is reported in section A.5. TB shows a striking ability to find all sorts of patterns in the data unaffected by artifacts and noise, as shown in section A.20. Another important experiment is to see how TB performs with high-dimensional data. For this purpose, we used 100 thousand text embeddings with 1024 dimensions from Hugging Face see details in section A.19. TB was able to achieve the highest scores while using the least amount of memory.

## 7 DISCUSSION

In this work, a new algorithmic set around Thetan Berserker (TB) was introduced. TB demonstrates an important advantage in terms of both speed and memory use. In addition, TB shows an increase in accuracy across many evaluation metrics, simulations, benchmarks, and real data experiments. Another surprising fact about TB is that it is able to provide highly accurate low-level segmentation. This is indeed surprising because we simply provided the location and intensity of each pixel as input. Usually, superpixel algorithms are specifically tailored for superpixel problems and highly optimized for that specific purpose. However, TB does not have this requirement nor was it built for that task in mind. Nonetheless, it generates useful superpixels at acceptable times. More importantly, the results match well the underlying images simplifying the underlying representations in a meaningful manner. Similarly, we achieve great overall segmentations for 3D imaging data. Both types of data are remarkably hard to process. Nonetheless, TB identifies the relevant structures in the data without compromising edges, which achieves an outstanding compression ratio (see Fig. 7). However, TB is also anticipated to struggle with the curse of dimensionality as any other clustering method. We were happy to report results in grouping text embedding of up to 1024 dimensions. TB is not bound by density limits in the same way that MeanShift, HDBSCAN, and DBSCAN do, as distributions become sparser when dimensions increase. It is also not bound to be affected by outliers as KMeans. TB outperformed KMeans++ seeding. TB is still aimed for linearly separable clusters and it will not perform well on nonlinearly separable clustering problems. In these cases, TBSCAN should be used instead (see Tab. A3). The proposed approach is general purpose and can be used across data science. More importantly, it provides some theoretical ground for the AI community to rethink and rework unsupervised learning. The code will be available on GitHub upon acceptance for publication.

## 8 CONCLUSION

Thetan Berserker (TB) is as uniquely accurate as a fast clustering algorithm. In addition, TB enables widely used algorithms by improving their conditioning and initialization. Important real-world applications were demonstrated, such as the segmentation of natural images, edge-preserving compression of magnetic resonance data, and grouping of text embeddings. TB outperformed more than 20 methods across 30 experiments.

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

# A APPENDIX

## A.1 APPARENT CENTROID METRIC

In this work, Apparent Centroid distance (AC) was introduced as a strict criterion to evaluate clustering results using the predicted centroids.

The metric is calculated as follows. The complete distance matrix between all estimated and ground truth centroids is created. We count the number of centroids close to the ground truth as being less than a pre-specified threshold. AC is bounded between 0 and 1. The recommended threshold is at 10% of the circumscribed radius of size $\theta$. 0.1 is used for all relevant experiments. AC drops quickly when even a small number of incorrect centroids are found. This is not the case with other metrics that were reported. This allows a quantifiable identification of issues that are rather easy for humans but hard for other metrics to catch. See Fig. A4-A15 and corresponding Tab. 1.

## A.2 HYPER-PARAMETER SELECTION AND SENSITIVITY

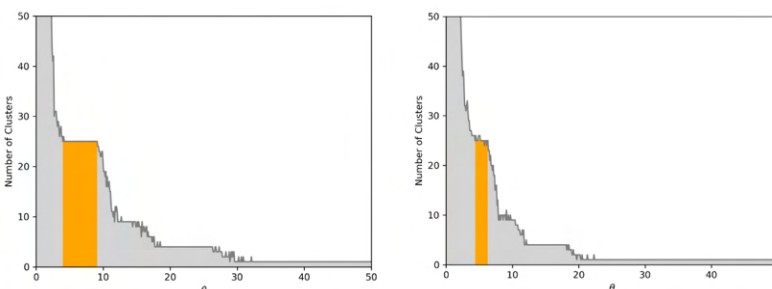

Figure A1: $\theta$ is often easier to find than $K$. The correct number of clusters and corresponding $\theta$ ranges were identified in the highlighted regions.

In this experiment, two-dimensional normal distributions of identity covariance matrices are added in a grid of either axes of dimensions -22 to 22 separated in 5 equal spaces (see Fig. A1-Left) or -15 to 15 at 5 equidistant spaces (see Fig. A1-Right). The left has clusters with more space in between them, and the right has less, i.e., clusters are closer. 100 points were sampled from each distribution in each grid position. Note that the true number of clusters is 25, and in both cases, it is easy to find just by looking at the plateaus of Fig. A1. Therefore, $\theta$ can be robust in areas capturing the correct number of clusters (see highlight). In summary, there are only 4-5 plateaus in both diagrams. The alternative would be to search $K$ in a brute-force manner. Such plots as that of Fig. A1 can be further improved by calculating the sum of intra-class differences. This allows us to even separate between the few plateaus. For example, the best plateau will also have the minimum sum of intra-class distances (also known as WCSS, within-cluster sum of squares).

## A.3 LARGE SIMULATION EXPERIMENT - PARAMETERS

The following parameters were used for each method of the large comparisons experiment. Most methods have many parameters. For parameters not reported it means that the default options were used. Gaussian Mixture Models (tied covariance, maxiter=200, 5 initializations) (Moon, 1996), TB ($\theta$ 3.6), TS ($\theta$ 3.6), ($\theta$ 3.6), TSR ($\theta$ 3.6, R 10), TBSCAN ($\theta$ 3.6, eps 2, min_samples 1), Bisecting KMeans (I 1, K 300), DBSCAN (eps 0.5 and min_samples 40), HDBSCAN (min_samples 40), KMeans++ (I 1, K 300), MeanShift (bandwidth 2.6), TBK ($\theta$ 3.6, K 300), FCM (n_clusters 300, expon 2, error 0.005, maxiter 1000) (Bezdek et al., 1984), MeanShift++ (bandwidth 3.4, threshold 0.00001, I 1000), OPTICS (min_samples 40, xi 0.05, min_cluster_size=5) (Ankerst et al., 1999), BSAS(theta 3.6, K 300), MBSAS (theta 3.6, K 300), TTSAS (threshold1 3.6, threshold2 4), BIRCH (threshold 2.5, branching_factor 500, n_clusters=K), CURE(n_clusters K, rep_points 1, compression 1) (Guha et al., 1998), KMedians (K 300) (Moshkovitz et al., 2020b), CLARANS (K 300, numlocal 6, maxneighbor 4) (Ng & Han, 2002). Note that KMedoids (Moshkovitz et al., 2020a), Affinity Propagation (Dueck, 2009), Spectral Clustering (Von Luxburg, 2007) and Hierarchical Clustering cannot run in this experiment due to extensive memory requirements (> 100 GBytes of RAM).

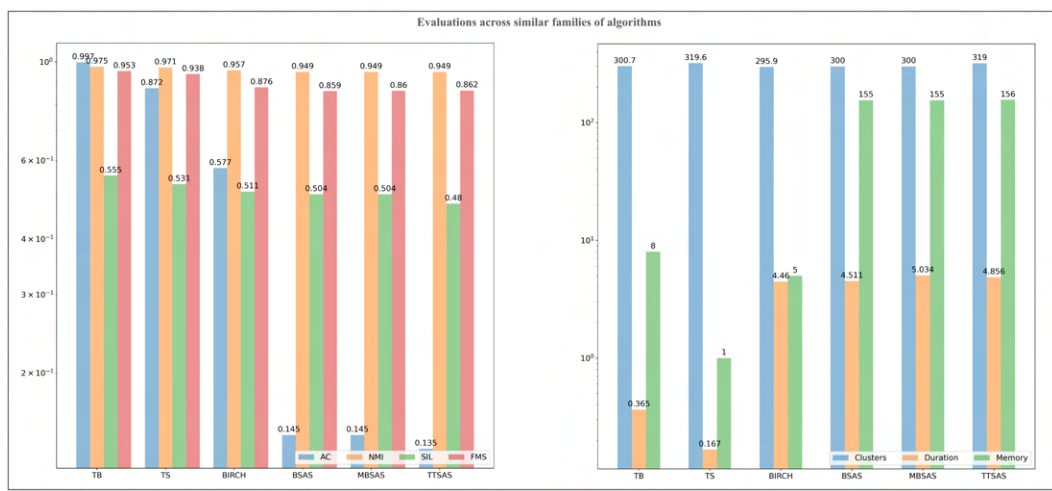

Figure A2: Summary plots with notable observations. Reporting AC, NMI, SIL, FMS, #clusters, runtime (duration in seconds), and peak memory (MBytes). Comparisons between families of algorithms that are distance-based and have common ground with TB. Note that TB has the highest scores while having the second fastest runtime after TS. For a complete list of comparisons, see Table 1.

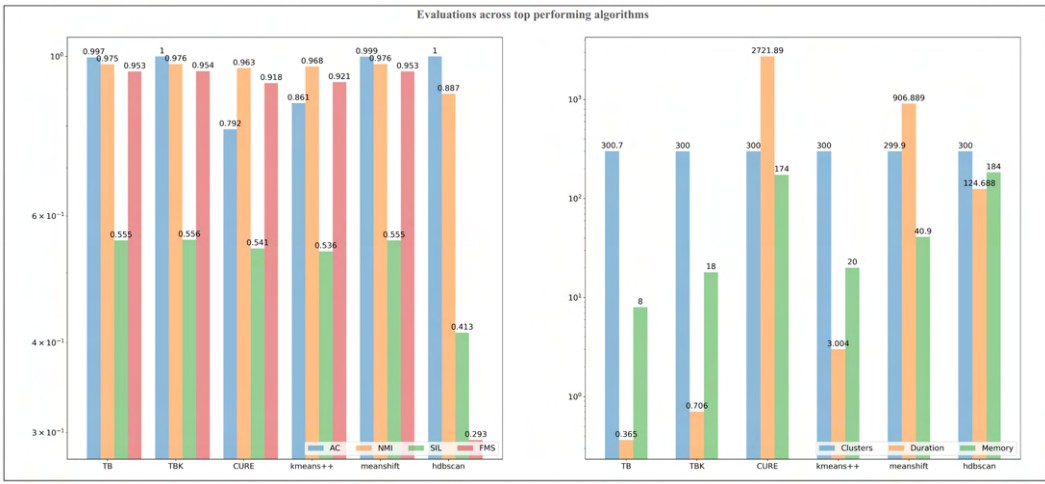

Figure A3: Comparisons between the 6 top performing algorithms in regards to AC. Note that TB matches the performance of algorithms while being 2484X faster and using 5X less memory than MeanShift and 342X faster and using 23X less memory than HDBSCAN. For a complete list of comparisons, see Table 1.

## A.4 LARGE SIMULATION EXPERIMENT - VISUALS

In support of the large comparisons experiment of section 5.1, we present visual results of a range of algorithms. Different colors encode different clusters. Blue crosses ground truth centroids, and red crosses encode estimated centroids (see Fig. A4-A15).

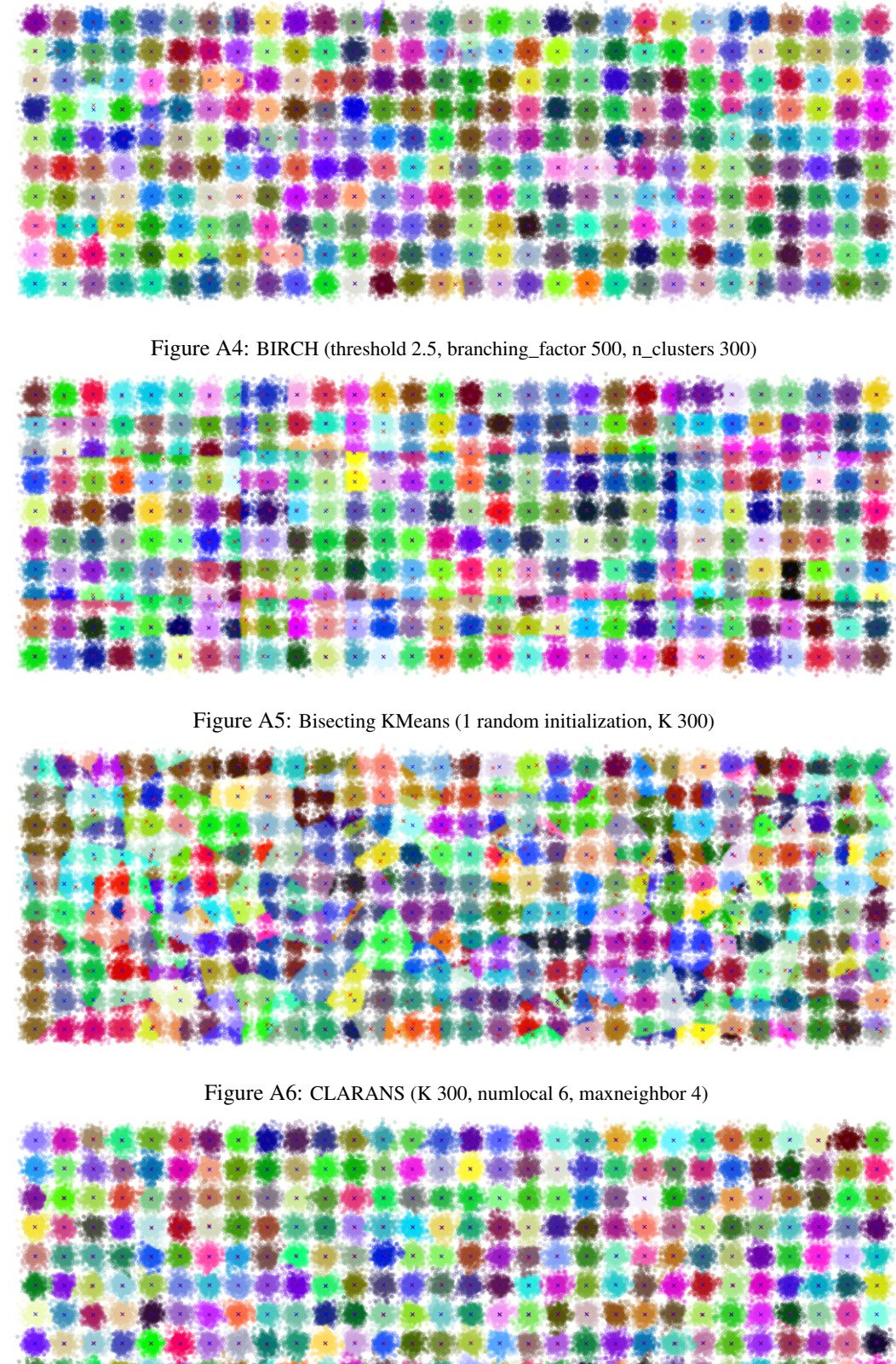

Figure A4: BIRCH (threshold 2.5, branching_factor 500, n_clusters 300)

Figure A5: Bisecting KMeans (1 random initialization, K 300)

Figure A6: CLARANS (K 300, numlocal 6, maxneighbor 4)

Figure A7: CURE (n_clusters 300, rep_points 1, compression 1)

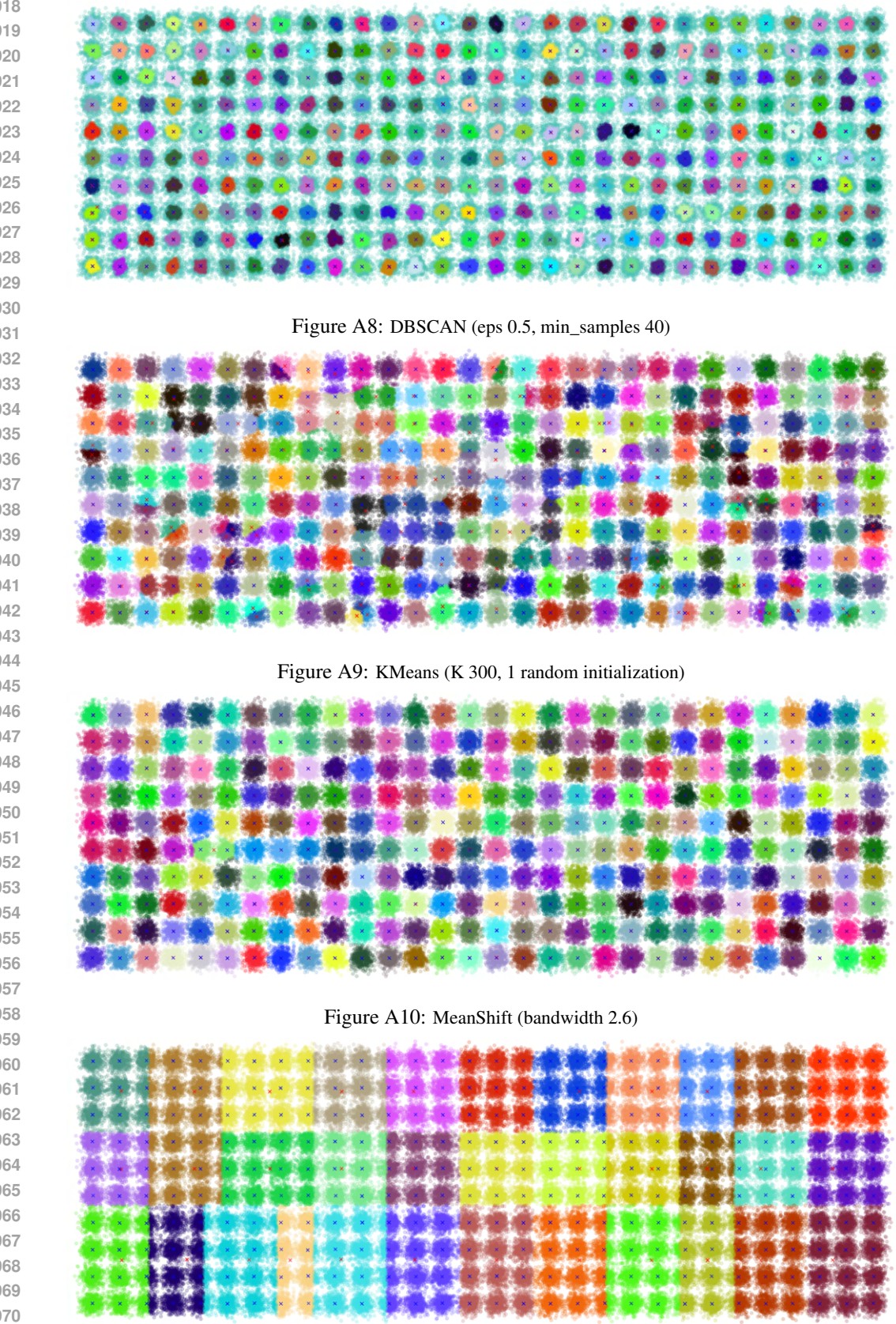

Figure A8: DBSCAN (eps 0.5, min_samples 40)

Figure A9: KMeans (K 300, 1 random initialization)

Figure A10: MeanShift (bandwidth 2.6)

Figure A11: MeanShift++ (bandwidth 3.4, threshold 0.00001, iterations 1000)

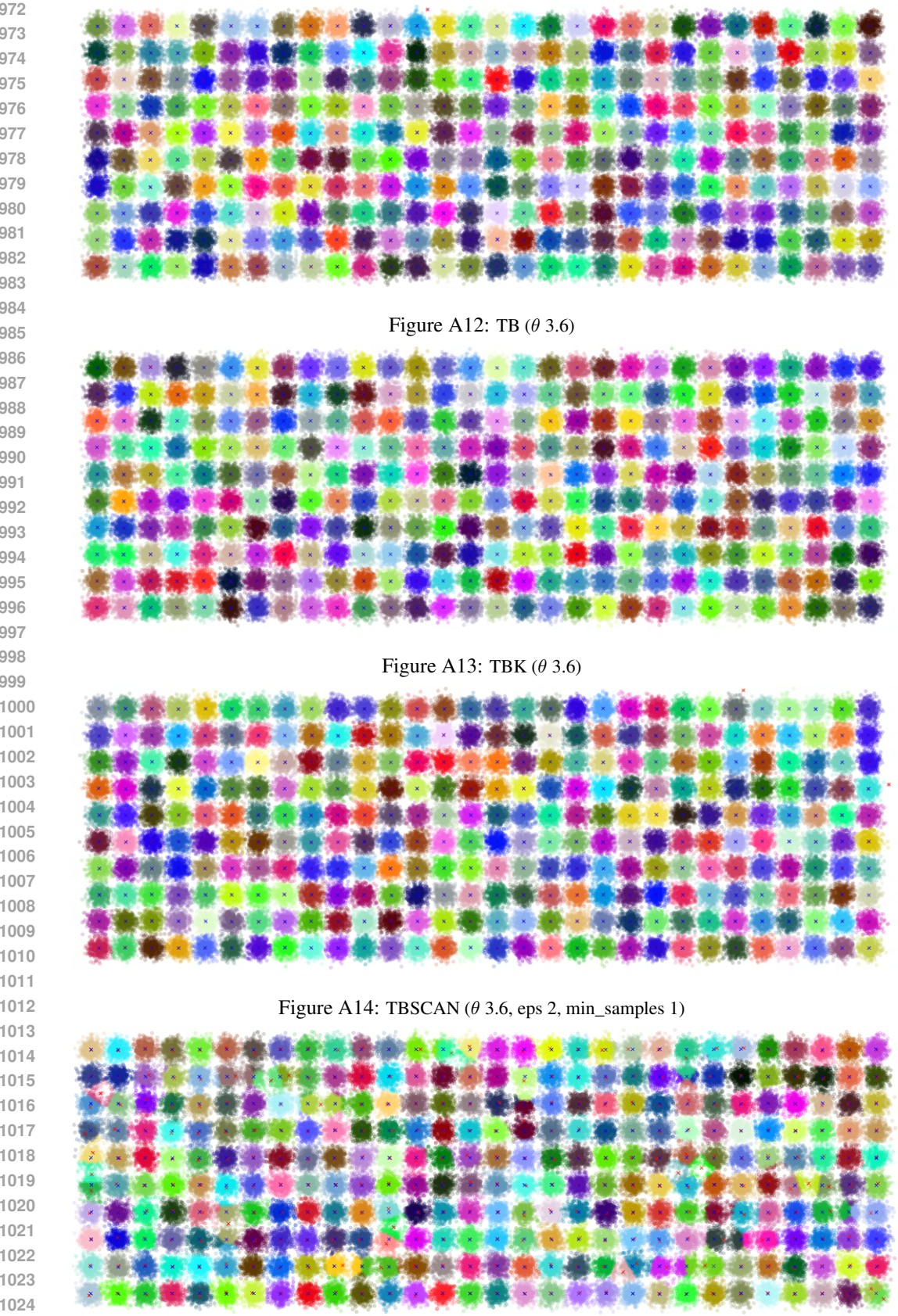

Figure A12: TB ($\theta$ 3.6)

Figure A13: TBK ($\theta$ 3.6)

Figure A14: TBSCAN ($\theta$ 3.6, eps 2, min_samples 1)

Figure A15: BSAS ($\theta$ 3.6, K 300)

## A.5 SIGNAL PROCESSING OF TIME SERIES

The ability of TB to identify patterns in one-dimensional signals that change over time is demonstrated here. For this purpose, common signals such as sine, pulses, or combined sine and pulse signals are used. In addition, real publicly available PTB-XL Electrocardiography (ECG) datasets (Wagner et al., 2020) are used. Fig. A16 to A23, show results obtained using TB. For Figs. A16 to A18, a small sliding window of 3, 5, and 3 is used, respectively. This sliding window becomes TB's feature space. The signal is normalized so that the two axes are relevant. Therefore, we move from a 1D signal to a 2D array $X$ where the number of rows is as many as the samples of the signal and the number of columns is the size of the sliding window. TB is then applied to this 2D array with $\theta$ values of 90, 130, and 95, respectively. Note that TB is identifying accurately the patterns in the signal. The colors correspond to different clustering labels.

For ECG Signals, two types of pre-processing were performed. In the first type of pre-processing, a Gaussian filter was used with $\sigma = 9$ (see Fig. A19), $\sigma = 6$ (see Fig. A20) and $\sigma = 11$ (see Fig. A21). Sliding windows of sizes 3, 7, and 3 were used, respectively. In the second type of pre-processing, i.e., for Fig. A22 and Fig. A23, the entire dataset is used along with time as the feature space. Sliding windows and Gaussian filter were not used in this method. The $\theta$ values that were used are 1, 6, 26, 90, and 50, respectively. The signal was normalized in both cases by scaling the data by 100. As shown in Fig. A16 to A23, TB with $L^2$ is able to identify the patterns in the signal without needing to calculate the derivatives (Duan & Guo, 2022). In the case of the PTB-XL ECG dataset, TB is able to identify the QRS and T-wave together (Fig. A19) or as QR, RS, and T-wave separately (Fig. A20) or as QRS and T-wave separately (Fig. A21) (Maglaveras et al., 1998). Alternatively, the full cardiac cycles are also captured as discrete groups (see Fig. A22 and Fig. A23).

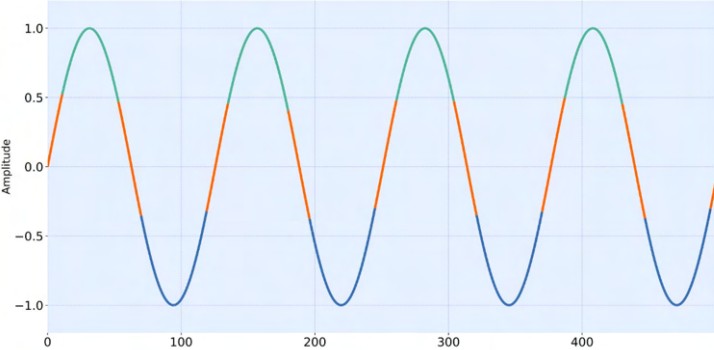

Figure A16: Patterns in Sine wave identified with TB.

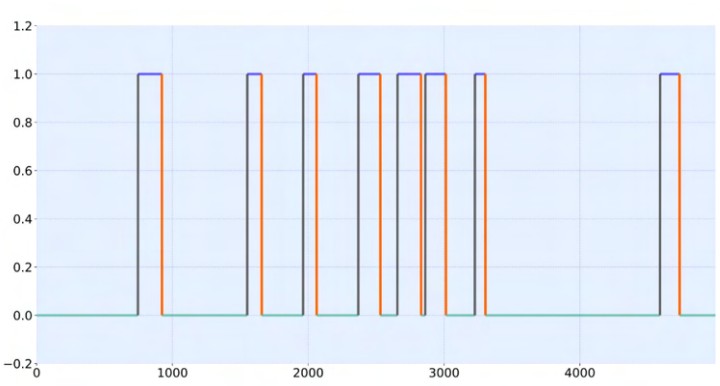

Figure A17: Patterns in Pulse signals identified with TB

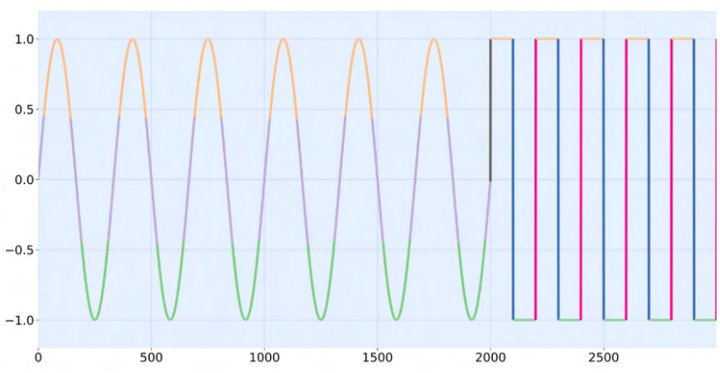

Figure A18: Patterns in Combined signal identified with TB

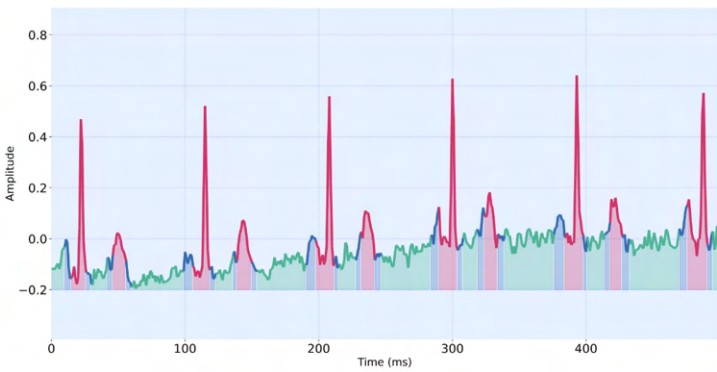

Figure A19: QRS and T-wave patterns identified together in ECG signals using TB

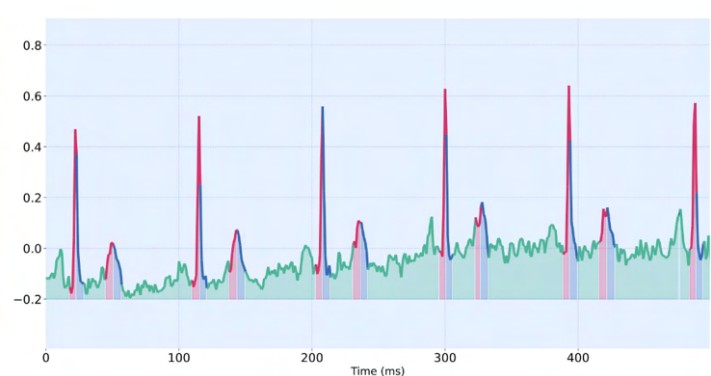

Figure A20: QR, RS, and T-wave patterns identified separately in ECG signals using TB

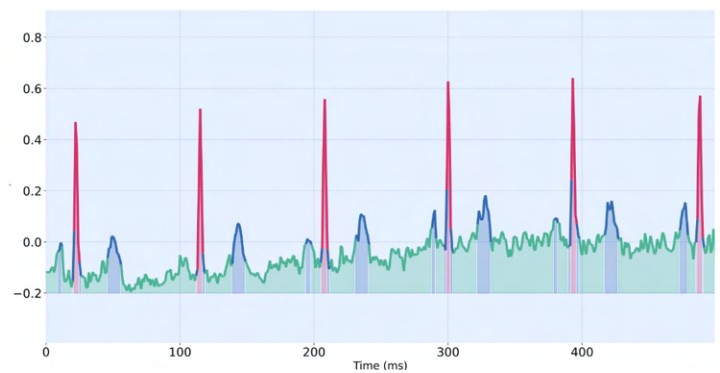

Figure A21: QRS and T-wave patterns identified together in ECG signals using TB

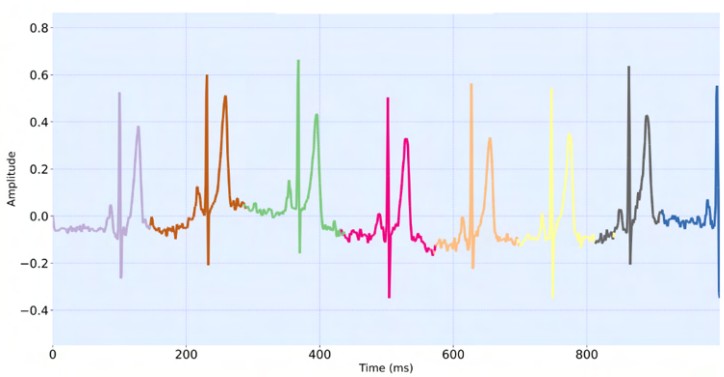

Figure A22: Full cardiac cycles in ECG signals identified with TB

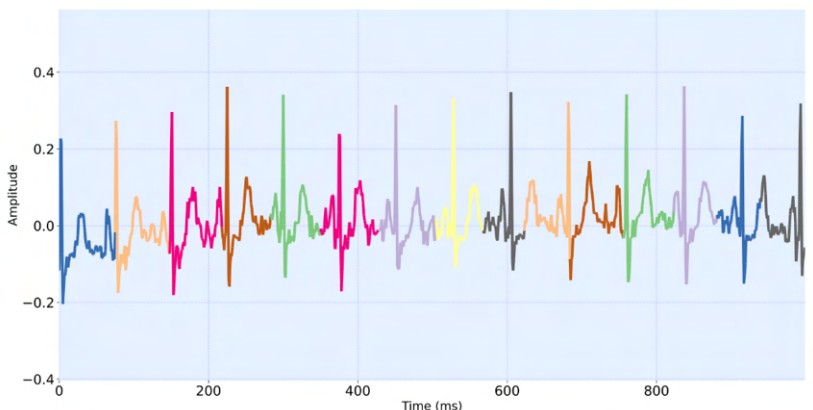

Figure A23: Full cardiac cycles in ECG signals identified with TB

## A.6 ADDITIONAL INFORMATION ON BENCHMARKS

The experimental setups are shown in Fig. A24. In addition, the official benchmarking scores are provided in Tab. A1, Tab. A2 and Tab. A3. The parameters used for the linearly and non-linearly separable datasets shown in section 5.2 are summarized in the tables below.

Tab. A1 demonstrates exceptional performance across all metrics (NCA, RI, FM, ARI, NMI) by TB method, achieving nearly perfect clustering results with remarkably low runtime compared to K-means, Birch, and Meanshift. TB consistently outperforms the other algorithms in efficiency, making it ideal for real-time or resource-constrained scenarios. However, TB may not inherently support non-linear separability, which could limit its application to datasets with more complex structures. TBSCAN has been introduced for this exact reason. As we can observe on table A2, TBSCAN stands out by maintaining excellent clustering performance (NCA, RI, FM, ARI, NMI), rivaling HDBSCAN and Agglomerative Clustering (single linkage), but with significantly lower runtime. A small note that other linkage type (ward, complete, average) has been tested with Agglomerative Clustering. However, we could not achieve excellent clustering so they had to be disclosed for a fair comparison.

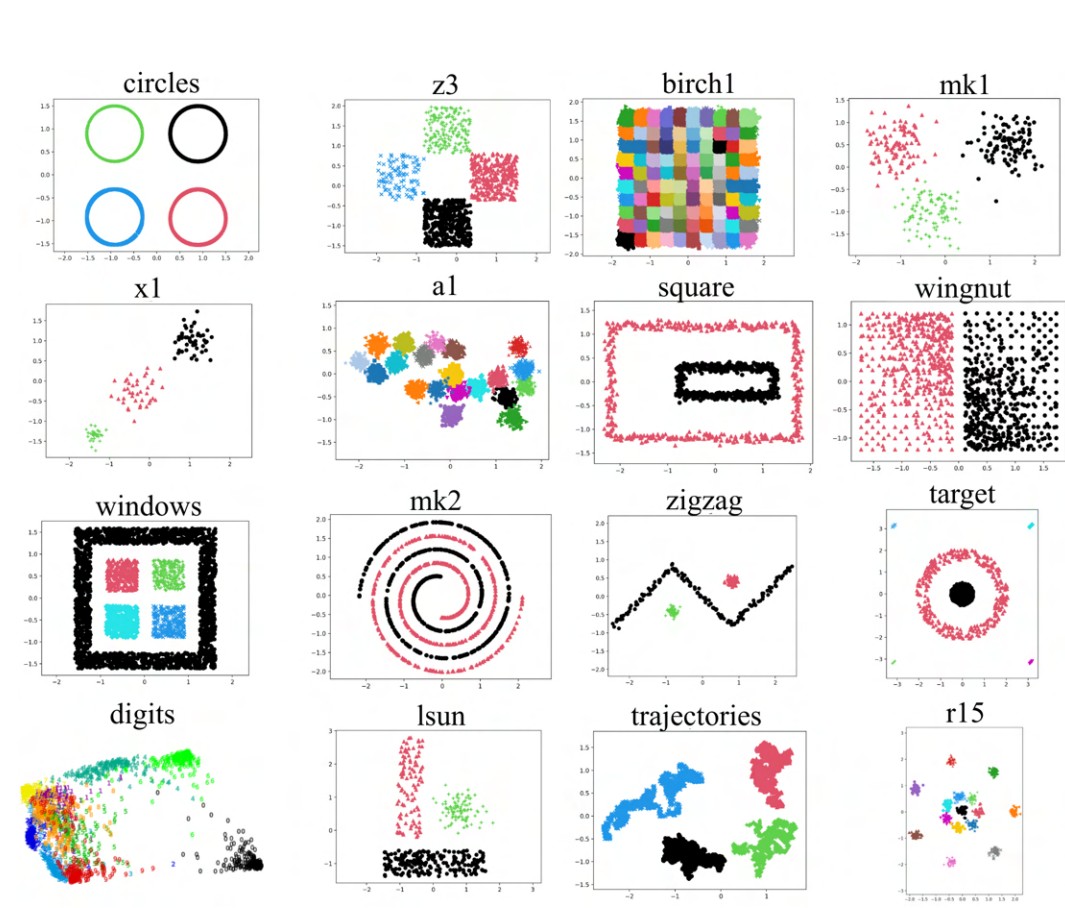

Figure A24: Linear and Non-Linear datasets used for the benchmark experiments

Table A1: Performance Comparisons of Linear benchmarks.

| Dataset | Clusters | Runtime ↓ | NCA ↑ | RI ↑ | FM ↑ | ARI ↑ | NMI ↑ |
|---|---|---|---|---|---|---|---|
| TB on Linearly Separable Datasets | | | | | | | |
| circles | 4/4 | **0.00091** | 1.0 | 1.0 | 1.0 | 1.0 | 1.0 |
| z3 | 4/4 | **0.00053** | 1.0 | 1.0 | 1.0 | 1.0 | 1.0 |
| birch1 | 101/100 | **0.07872** | 0.97 | 0.99 | 0.97 | 0.96 | 0.98 |
| mk1 | 3/3 | **0.00017** | 0.99 | 0.99 | 0.99 | 0.98 | 0.98 |
| trajectories | 4/4 | **0.00219** | 0.99 | 0.99 | 0.99 | 0.99 | 0.99 |
| x1 | 3/3 | **0.00013** | 1.0 | 1.0 | 1.0 | 1.0 | 1.0 |
| a1 | 24/20 | **0.00091** | 0.81 | 0.99 | 0.99 | 0.95 | 0.96 |
| r15 | 8/8 | **0.00031** | 1.0 | 1.0 | 1.0 | 1.0 | 1.0 |
| K-means on Linearly Separable Datasets | | | | | | | |
| circles | 4/4 | 0.00634179 | 1.0 | 1.0 | 1.0 | 1.0 | 1.0 |
| z3 | 4/4 | 0.00363559 | 1.0 | 1.0 | 1.0 | 1.0 | 1.0 |
| birch1 | 100/100 | 0.53998762 | 0.95681 | 0.99888 | 0.96278 | 0.94378 | 0.97568 |
| mk1 | 3/3 | 0.00116613 | 0.99502 | 0.99556 | 0.99668 | 0.98998 | 0.98299 |
| trajectories | 4/4 | 0.01502252 | 0.99987 | 0.99990 | 0.99993 | 0.99973 | 0.99936 |
| x1 | 3/3 | 0.00089175 | 1.0 | 1.0 | 1.0 | 1.0 | 1.0 |
| a1 | 25/20 | 0.00624223 | 0.98282 | 0.99682 | 0.99833 | 0.96634 | 0.97381 |
| r15 | 8/8 | 0.00212648 | 1.0 | 1.0 | 1.0 | 1.0 | 1.0 |
| Birch on Linearly Separable Datasets | | | | | | | |
| circles | 4/4 | 0.01960280 | 1.0 | 1.0 | 1.0 | 1.0 | 1.0 |
| z3 | 4/4 | 0.01141702 | 1.0 | 1.0 | 1.0 | 1.0 | 1.0 |
| birch1 | 100/100 | 1.69575003 | 0.89768 | 0.91556 | 0.9602470 | 0.95321 | 0.966945 |
| mk1 | 3/3 | 0.00366206 | 0.99502 | 0.99556 | 0.99668 | 0.98998 | 0.98299 |
| trajectories | 4/4 | 0.04717597 | 1.0 | 1.0 | 1.0 | 1.0 | 1.0 |
| x1 | 3/3 | 0.0028004 | 1.0 | 1.0 | 1.0 | 1.0 | 1.0 |
| a1 | 23/20 | 0.0196028 | 0.79705 | 0.99222 | 0.99591 | 0.91804 | 0.95122 |
| r15 | 8/8 | 0.00667788 | 1.0 | 1.0 | 1.0 | 1.0 | 1.0 |
| Meanshift on Linearly Separable Datasets | | | | | | | |
| circles | 4/4 | 2.64446 | 1.0 | 1.0 | 1.0 | 1.0 | 1.0 |
| z3 | 4/4 | 1.54018 | 1.0 | 1.0 | 1.0 | 1.0 | 1.0 |
| birch1 | 101/100 | 228.76032 | 0.96543 | 0.981043 | 0.960004 | 0.957838 | 0.9785959 |
| mk1 | 3/3 | 0.49402 | 0.99 | 0.99 | 0.99 | 0.98 | 0.98 |
| trajectories | 4/4 | 6.36414 | 0.99 | 0.99 | 0.99 | 0.99 | 0.99 |
| x1 | 3/3 | 0.37778 | 1.0 | 1.0 | 1.0 | 1.0 | 1.0 |
| a1 | 25/20 | 2.64446 | 0.86 | 0.98 | 0.97 | 0.96 | 0.97 |
| r15 | 8/8 | 0.90086 | 1.0 | 1.0 | 1.0 | 1.0 | 1.0 |

Table A2: Performance Comparisons of Non-Linear benchmarks.

| | | TBSCAN on Non-Linearly Separable Datasets | | | | |
|---|---|---|---|---|---|---|
| Dataset | Clusters | Runtime ↓ | NCA ↑ | RI ↑ | FM ↑ | ARI ↑ | NMI ↑ |
| windows | 5/5 | **0.007092875** | 1.0 | 1.0 | 1.0 | 1.0 | 1.0 |
| square | 2/2 | **0.001162792** | 1.0 | 1.0 | 1.0 | 1.0 | 1.0 |
| wingnut | 2/2 | **0.001298250** | 1.0 | 1.0 | 1.0 | 1.0 | 1.0 |
| zigzag | 3/3 | **0.000820125** | 1.0 | 1.0 | 1.0 | 1.0 | 1.0 |
| target | 6/6 | **0.002047375** | 1.0 | 1.0 | 1.0 | 1.0 | 1.0 |
| lsun | 3/3 | **0.001416042** | 1.0 | 1.0 | 1.0 | 1.0 | 1.0 |
| mk2 | 2/2 | **0.003112458** | 1.0 | 1.0 | 1.0 | 1.0 | 1.0 |

| | | HDBSCAN on Non-Linearly Separable Datasets | | | | |
|---|---|---|---|---|---|---|
| Dataset | Clusters | Runtime ↓ | NCA ↑ | RI ↑ | FM ↑ | ARI ↑ | NMI ↑ |
| windows | 5/5 | 0.038237875 | 1.0 | 1.0 | 1.0 | 1.0 | 1.0 |
| square | 2/2 | 0.016157084 | 1.0 | 1.0 | 1.0 | 1.0 | 1.0 |
| wingnut | 2/2 | 0.019175417 | 1.0 | 1.0 | 1.0 | 1.0 | 1.0 |
| zigzag | 3/3 | 0.002936958 | 1.0 | 1.0 | 1.0 | 1.0 | 1.0 |
| target | 3/6 | 0.006192208 | 0.50000 | 0.99982 | 0.99982 | 0.99963 | 0.98602 |
| lsun | 3/3 | 0.004924541 | 1.0 | 1.0 | 1.0 | 1.0 | 1.0 |
| mk2 | 2/2 | 0.017242167 | 1.0 | 1.0 | 1.0 | 1.0 | 1.0 |

| | | Agglomerative Clustering (Single Linkage) on Non-Linearly Separable Datasets | | | | |
|---|---|---|---|---|---|---|
| Dataset | Clusters | Runtime ↓ | NCA ↑ | RI ↑ | FM ↑ | ARI ↑ | NMI ↑ |
| windows | 5/5 | 0.029719333 | 1.0 | 1.0 | 1.0 | 1.0 | 1.0 |
| square | 2/2 | 0.005899709 | 1.0 | 1.0 | 1.0 | 1.0 | 1.0 |
| wingnut | 2/2 | 0.006111000 | 1.0 | 1.0 | 1.0 | 1.0 | 1.0 |
| zigzag | 3/3 | 0.001121333 | 1.0 | 1.0 | 1.0 | 1.0 | 1.0 |
| target | 6/6 | 0.004177458 | 1.0 | 1.0 | 1.0 | 1.0 | 1.0 |
| lsun | 3/3 | 0.002066667 | 1.0 | 1.0 | 1.0 | 1.0 | 1.0 |
| mk2 | 2/2 | 0.005654750 | 1.0 | 1.0 | 1.0 | 1.0 | 1.0 |

| | | Spectral Clustering on Non-Linearly Separable Datasets | | | | |
|---|---|---|---|---|---|---|
| Dataset | Clusters | Runtime ↓ | NCA ↑ | RI ↑ | FM ↑ | ARI ↑ | NMI ↑ |
| windows | 5/5 | 3.573450375 | 0.76499 | 0.56371 | 0.68490 | 0.08047 | 0.36880 |
| square | 2/2 | 0.568301875 | 0.61350 | 0.56802 | 0.47644 | 0.13638 | 0.25496 |
| wingnut | 2/2 | 0.374673375 | 0.86220 | 0.87157 | 0.87170 | 0.74314 | 0.63820 |
| zigzag | 3/3 | 0.023156958 | 0.47152 | 0.70699 | 0.80126 | 0.30452 | 0.46853 |
| target | 6/6 | 0.502081875 | 0.28023 | 0.69488 | 0.67860 | 0.39250 | 0.48070 |
| lsun | 3/3 | 0.296371333 | 0.87513 | 0.89550 | 0.91390 | 0.78268 | 0.80938 |
| mk2 | 2/2 | 0.460497625 | 0.08236 | 0.50286 | 0.50228 | 0.00573 | 0.00488 |

Table A3: Performance Comparisons on Digits dataset between TB and HC.

| Methods | Clusters | Runtime ↓ | Peak Memory ↓ | NCA ↑ | RI ↑ | FM ↑ | ARI ↑ | NMI ↑ |
|---|---|---|---|---|---|---|---|---|
| HC-Ward | 10/10 | 0.028286958 | 13.8585 | **0.5896** | **0.9049** | **0.9467** | **0.5129** | **0.6291** |
| HC-Average | 10/10 | 0.027868458 | 13.8541 | 0.5383 | 0.8526 | 0.9151 | 0.4179 | 0.6139 |
| HC-Complete | 10/10 | 0.025582417 | 13.8534 | 0.4858 | 0.8589 | 0.9194 | 0.3844 | 0.5702 |
| HC-Single | 10/10 | 0.014241500 | 13.8534 | 0.1960 | 0.2801 | 0.4478 | 0.0447 | 0.2418 |
| TB | 10/10 | **0.000827417** | **0.1469** | 0.5222 | 0.8425 | 0.9092 | 0.3749 | 0.5752 |

Table A4: TB/TBSCAN Parameters for Linearly/Non-Linearly Separable Datasets experiments

| Linear Dataset | $\theta$ | Non-Linear Dataset | $\theta$ | Min Samples | EPS |
|---|---|---|---|---|---|
| circles | 1.2 | trajectories | 0.1 | 1 | 0.3 |
| z3 | 1.1 | windows | 0.1 | 1 | 0.3 |
| birch1 | 0.25 | square | 0.2 | 1 | 0.4 |
| mk1 | 1.5 | wingnut | 0.25 | 1 | 0.42 |
| unbalance | 0.35 | zigzag | 0.25 | 1 | 0.5 |
| x1 | 1 | target | 0.1 | 1 | 0.5 |
| a1 | 0.3 | lsun | 0.1 | 1 | 0.35 |
| r15 | 1 | mk2 | 0.05 | 1 | 0.35 |

## A.7 ADDITIONAL SUPERPIXEL EXPERIMENTS

Additional information on superpixel segmentation is provided here. Fig. A25 compares TB against KMeans++, MeanShift and MeanShift++. Superpixel labels and the averaged images are shown using two examples from the BSD500 (first and second row) and NYUV2 dataset (third and fourth row).

Fig. A26 presents how the change in number of superpixels (or clusters) affects established superpixel segmentation scores. Here, SLIC with different parameters is compared against TB, MeanShift, and MeanShift++. KMeans++ was excluded from this experiment due to the long runtime of the method. The metrics were evaluated on the BSD500 dataset. Since the dataset contains multiple labels per image, the metrics were first calculated for all labels of the image and averaged. The value on the plot shows the average of such values from 500 images of the dataset.

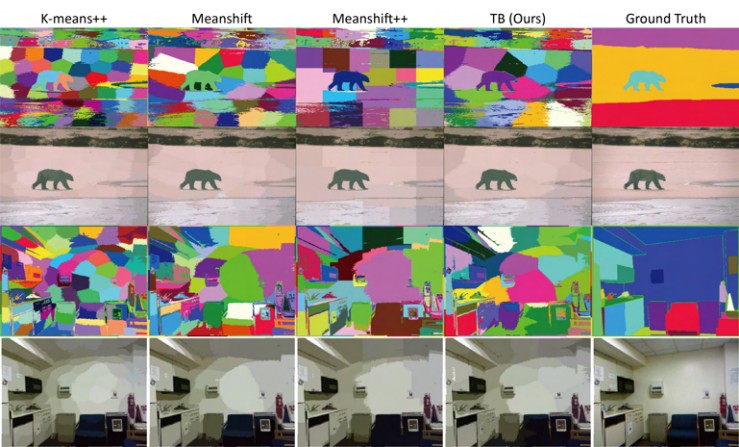

Figure A25: Superpixel comparison against other clustering methods

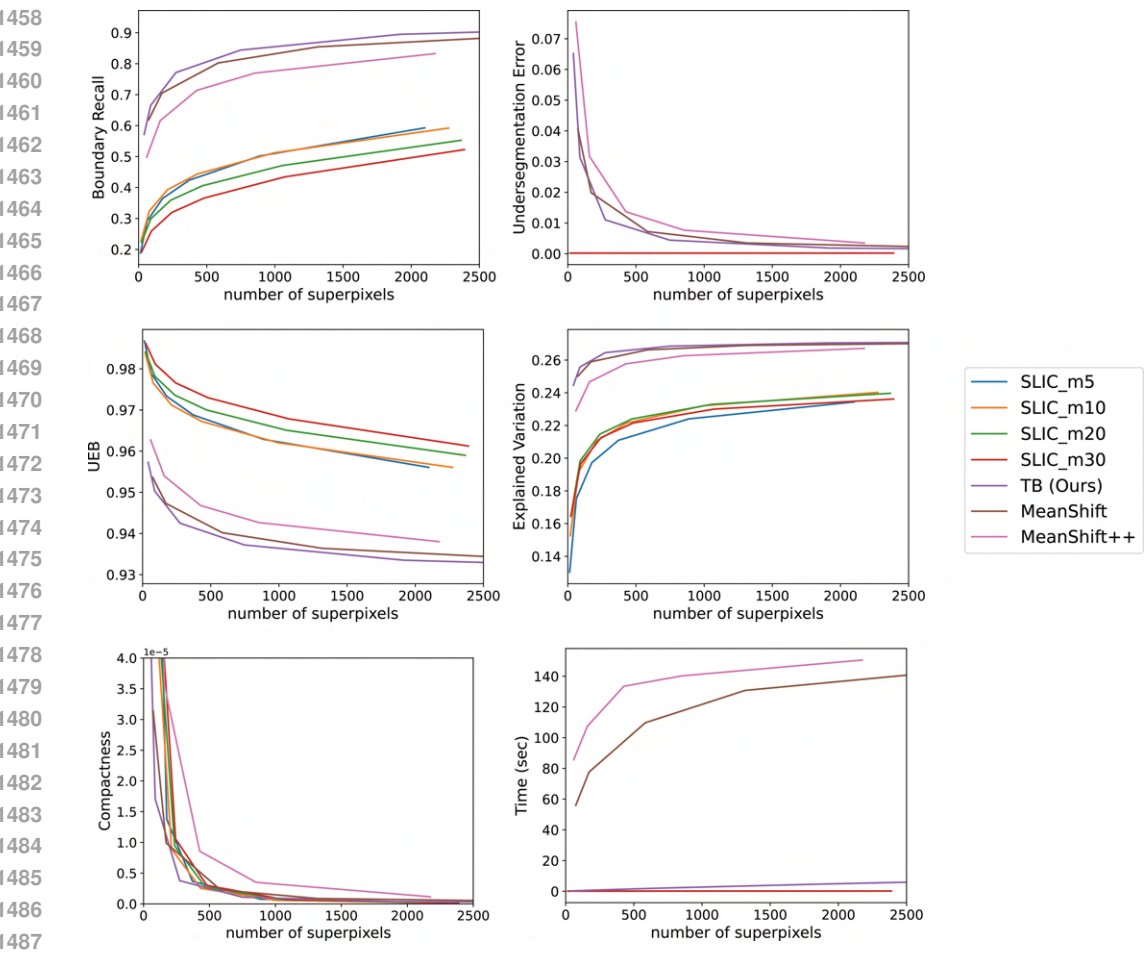

Figure A26: The plot shows how methods perform on superpixel tasks across different metrics.

## A.8 PROVING RUNTIME

**Theorem 3** The runtime complexity of Thetan Berseker (TB) in regards to distance calls is in the range of O $(n)$ to O $(n^2)$.

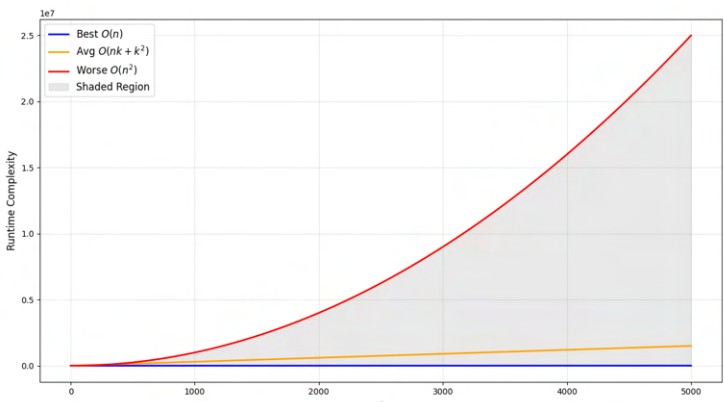

Figure A27: Thetan Berserker's runtime complexity is bounded between $\mathcal{O}(n)$ and $\mathcal{O}(n^2)$.

**Proof** Thetan Berserker (TB) consists of 4 steps as described in the ablation study (see Fig. 3A). Two of the steps run Thetan Sequential (TS steps), and two-run centroid updates and relabeling (CL

steps). We define $N$ as the total number of samples and $K$ as the total number of clusters. In order to connect to the previous definitions, we will use $T(N)/O(N)$ rather than $T(n)/O(n)$. In addition, we will omit $D$ (feature space size) for now, given that we look at complexity in regard to distance calls.

The worst Big-O time complexity for Thetan Sequential is O ($N^2$). This is because most of the compute time is spent on distance computations. Therefore in the case where each sample is a different cluster, we obtain a total of $1 + 2 + 3 + ... + N - 1 = N(N-1)/2 = N^2/2 - N/2$ distances. This will run in length two times for TB.

The second time, the centroids are pre-pended to the sample datasets; therefore, the total number of distance computations will be $1 + 2 + 3 + ... + N^-1$.

The CL steps of TB allow the merging of centroids and update labels. The first part of CL involves running TS only on the $M$ generated centroids. This will merge any centroids that have a neighbor centroid at a distance closer than $\theta$. The relabel step (RE) will update the final labels to account for the centroid updates. There are as many labels as the number of samples $N$. However, unique labels are only $M$. Putting all these together, we can study the best, worst, and average time complexity.

In the worst case, every sample is a singleton cluster, and for this reason, $K = N$. We can now separate the distance calls for each step of the algorithm.

Worst case $K = N$,

$$
T(N) = 
\begin{array}{ll}
\left.\begin{array}{l}
1 + 2 + \cdots + (N-1)] \quad TS_1(X) \\
+ 1 + 2 + \cdots + (N-1)] \quad \left.\begin{array}{l} TS_1(M) \\ \end{array}\right] \\
+ 0 \qquad\qquad\qquad\qquad ] \quad RE_1(N)
\end{array}\right] CL_1 \\
\left.\begin{array}{l}
+ 1 + 2 + \cdots + (N-1)] \\
+ N + N + \cdots + N
\end{array}\right] TS_2(M+X) \\
\left.\begin{array}{l}
+ 1 + 2 + \cdots + (N-1)] \quad TS_2(M) \\
+ 0 \qquad\qquad\qquad\qquad ] \quad RE_2(N)
\end{array}\right] CL_2
\end{array}
$$

Because there is nothing to relabel, the number of distance calls is 0. Therefore $T(N) = 4N^2/2 - 4N/2 + N^2 = 3N^2 - 2N$ which means that worst case Big-O for runtime is O ($N^2$). The best case takes place when the data is represented by a single cluster. Best case $K = 1$,

$$
T(N) = 
\begin{array}{ll}
\left.\begin{array}{l}
1 + 1 + \cdots + 1] \quad TS_1(X) \\
+ 1 + 1 + \cdots + 1] \quad \left.\begin{array}{l} TS_1(M) \\ \end{array}\right] \\
+ 0 \qquad\qquad\quad ] \quad RE_1(N)
\end{array}\right] CL_1 \\
\left.\begin{array}{l}
+ 1 + 1 + \cdots + 1] \\
+ 1 + 1 + \cdots + 1
\end{array}\right] TS_2(M+X) \\
\left.\begin{array}{l}
+ 1 + 1 + \cdots + 1] \quad TS_2(M) \\
+ 0 \qquad\qquad\quad ] \quad RE_2(N)
\end{array}\right] CL_2
\end{array}
$$

There is also no need for relabeling here as long as no new clusters are created. In other words, $T(N) = 5N$, which means that the best case has complexity O ($N$).

We should also look at the average case. In most clustering problems, the number of clusters is at least one order or many orders of magnitude less than the data samples. For example, for 1 million samples is not uncommon to search for 1 thousand clusters. Therefore, it is reasonable to assume that for an average case, $K$ is assumed to be smaller than $N$.

Average case, for $K << N$,

$$
T(N) = 
\begin{array}{ll}
1 + 2 + \cdots + (K-1) + (N-K-1)K] & TS_1(X) \\
+1 + 2 + \cdots + (K-1) & ]\ TS_1(M) \\
+1 + 2 + \cdots + (K-1) & ]\ RE_1(N) \\
+1 + 2 + \cdots + (K-1) + NK & ]\ TS_2(M+X) \\
+1 + 2 + \cdots + (K-1) & ]\ TS_2(M) \\
+1 + 2 + \cdots + (K-1) & ]\ RE_2(N)
\end{array}
$$

Which is equal to $T(N) = 2NK + 2.5K^2 - K - 3.5$. Therefore, the average case is O $(NK + K^2)$. Given that K is much less than N the runtime will be closer to linear than quadratic. This completes the proof. ∎

High dimensional data with a large number of features $D$ are expected to delay each distance computation in a constant matter. TB is used with $I = 2$ everywhere in this work. Given its fast convergence, as shown in Fig. 3B, we do not expect to see any surprises in regard to time complexity.

### A.9 PROVING MEMORY

**Lemma 3** The spatial complexity of Thetan Berserker is linear.

**Proof** The only memory generated is centroids and labels. Those are produced in a constant amount. Therefore, the spatial complexity is best case $T(N) = cN$, i.e. $\mathcal{O}(N)$ and worst case i.e. $T(N) = 2cN$, i.e. $\mathcal{O}(N)$. Where $c$ is a constant. ∎

### A.10 MERGING SAMPLES VIA THETAN SEQUENTIAL

Clarifications on Algorithm 2. As Theorem 1 suggests, TS works well when the clusters have inter-class distances greater than hyper-parameter $\theta$. In other words, when there is plenty of empty space between the clusters. However, the first TS in TB generates centroids. These according to Theorem 2 are reduced representation of the original data. Due to TS being order-sensitive, it could be that some of the centroids might be closer than $theta$. Nonetheless, they will be sparser than the original data if a reasonable $\theta$ has been chosen. Therefore, the second TS that acts on the centroids (output of the first TS) will merge together any centroids that are closer than $\theta$. See Fig. A28.

**Proposition 1** The output centroids of TS of hyper-parameter $\theta$ will be merged by a second TS on the centroids only if the centroids have in-between distances less than $\theta$.

**Proof** Due to Theorem 1, any order of selection will lead to singleton clusters of points that are far from each other by a distance that is greater than $\theta$. TS will generate such points, but due to its order sensitivity, it may generate a few centroids that are also close to each other. Therefore, we have a scenario such as that of Fig. A28. In that case, in order to prove this, we need to look at all pairwise distances and order selections. For example, if TS processes points in this order 1, 2, 3, 4. The distances between 1 and 2 are greater than the threshold (2 clusters), 2 and 3 are less than the threshold (2 and 3 get merged), and 3, and 4 are greater than the threshold (a new cluster). If we take any other ordering, for example 3, 1, 2, 4 we will conclude on the same result. Merging of 2 & 3. This is because the only small pairwise distance is between 2 and 3. This generalizes to any number of TS output centroids. ∎

Overall, this is an efficient way to merge close points without calculating all pairwise distances.

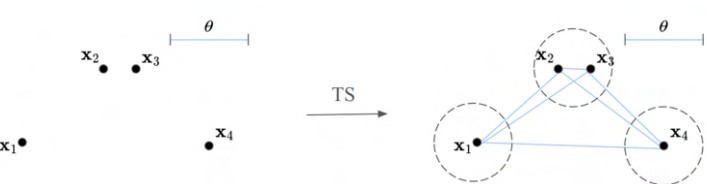

Figure A28: If distance $\theta$ has the size shown above, any order of selection of the points will allow TS (Algorithm 1) to merge samples 2 and 3.

### A.11 EFFICIENTLY UPDATING CENTROIDS

The centroids are updated on the fly using the following idea. The sum of samples for each cluster and the number of clusters are kept as different variables. To insert a new point $x_i$ to a cluster, we simply need to $\sum_k = \sum_k +x_i$ for each dimension $D$ and then update $n_k = n_k + 1$. The centroid evaluation is then performed only when needed by dividing the sums by their corresponding $n_k$.

### A.12 FURTHER CLARIFICATIONS ON TB'S FAST CONVERGENCE

This section is expanding on the proof of Theorem 2. In Fig. A29, we see two square regions representing two uniform distributions that are close to each other by distance $l$. In the diagram, $l$ is set to be at $\theta/2$.

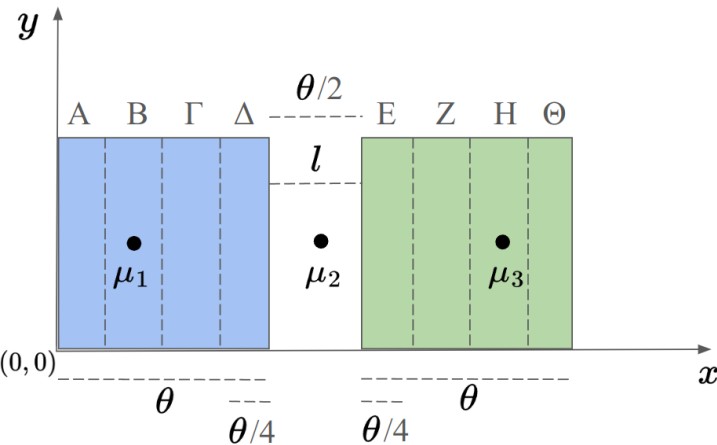

Figure A29: Two square uniform distributions (blue and green) of size $\theta \times \theta$ containing an undetermined number of samples are at a distance $l = \theta/2$. Greek letters separate the squares in equal bands.

Let $\mu_1$, $\mu_2$, and $\mu_3$ represent undesired centroids obtained from the initial clustering process using TS (due to a bad order). Upon applying a further clustering process, i.e, TS $(M, X)$, we encounter three possible cases:

    **Case 1:** If the distance between $\mu_1$ and $\mu_2$ is less than the threshold $\theta$, this results in two clusters:

        – **Cluster 1** has a new centroid at position: $(\mu_1 + \mu_2)/2$
        – **Cluster 2** has an updated centroid at position: $\mu_3'$

    **Case 2:** If the distance between $\mu_2$, and $\mu_3$ is less than $\theta$, $\mu_1$ is far from $\mu_2$, this results in two clusters:

        – **Cluster 1** has an updated centroid at position: $\mu_1'$
        – **Cluster 2** has a new centroid at position: $(\mu_2 + \mu_3)/2$

    **Case 3:** If all in between distances are greater than $\theta$ then 3 clusters are obtained:

        – **Cluster 1** has centroid at position: $\mu_1$
        – **Cluster 2** has centroid at position: $\mu_2$
        – **Cluster 3** has centroid at position: $\mu_3$

The index here corresponds to the actual order. Meaning that $\mu_1$ is the first point to be processed by TS$(M, X)$. $\mu_2$ is the second etc. It is important to note that these cases assume the distances between centroids are calculated using Euclidean distance.

The first two cases are highly desirable because they will help TB keep the two clusters separate and never provide three clusters. Therefore, cases 1 and 2 clearly improve order sensitivity. Even if an upcoming point is at the edge of each distribution, it will be pulled to the correct centroids and assigned to the correct cluster.

The third case is problematic because it may continue supporting the idea that there may be three (incorrect) rather than two clusters (clusters). We can prove by contradiction that case 3 is highly unlikely. This also further explains why TB converges as fast as it does.

**Proposition 2** Case 3 is not possible with the current setup.

Each uniform distribution (square) has $\theta$ sides. Let's assume that the origin of the coordinate system is at the bottom left corner of the blue square. Then $\boldsymbol{\mu}_2 = (\theta + \theta/4, \theta/2)$. Let's also assume that all three samples are at the same height ($y$ value). Therefore we can reduce this to look only at the $x$ axis.

As discussed in case 3, the distance between $\mu_1^x$ and $\mu_2^x$ is greater than $\theta$. Therefore,

$$\begin{aligned}
||\mu_2^x - \mu_1^x|| &> \theta \\
\theta + \theta/4 - \mu_1^x &> \theta \\
\mu_1^x &< \theta/4
\end{aligned}$$

Also the distance between $\mu_2^x$ and $\mu_2^x$ is greater than $\theta$, which means that

$$\begin{aligned}
||\mu_3^x - \mu_2^x|| &> \theta \\
\mu_3^x - \theta - \theta/4 &> \theta \\
\mu_3^x &> 2\theta + \theta/4
\end{aligned}$$

So if 3 centroids appear (rather than 2), two of them ($\mu_1^x$ and $\mu_3^x$) will be forced to be at the bands A and $\Theta$ respectively. But if this is the case, then $\mu_2^x$ will be representing bands B, $\Gamma$, $\Delta$, E, Z, H and the empty space of width $\theta/2$. But this is a contradiction because the threshold is only $\theta$, not $7\theta/2$. In other words there is no ordering that can allow TS to generate centroids that will be at such distances apart. This completes the proof. ∎

Here, we showed this contradiction with two clusters, but it is trivial to show the same for any number of uniform densities at $\theta/2$. The example shown here cannot cover all possible datasets, but it gives a good idea of why the Berserker centroids help create more accurate upcoming centroids.

A.13 USING RANDOM WALKS FOR PREDICTING THE HYPER-PARAMETER

The approach in regards to Fig. 4A is elaborated in this section. A random walk, in this case, is accessing random samples from the available data matrix $\boldsymbol{X}$.

For example, one random walk will visit samples $\boldsymbol{x}_0, \boldsymbol{x}_{10}, \boldsymbol{x}_{12}$ and $\boldsymbol{x}_{35}$. Another random will visit samples $\boldsymbol{x}_{30}, \boldsymbol{x}_{11}, \boldsymbol{x}_2$ and $\boldsymbol{x}_0$. The same number of jumps for the two random walks is kept.

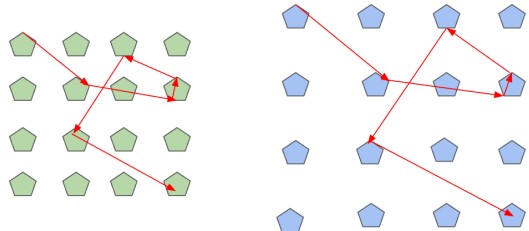

Figure A30: The lengths of the vectors of the same random walk can change as inter-cluster distances change.

Next, the lengths of the difference vectors between subsequent jumps are calculated. For example, from the first random walk we keep $||\boldsymbol{x}_{10} - \boldsymbol{x}_0||, ||\boldsymbol{x}_{12} - \boldsymbol{x}_{10}||$ and $||\boldsymbol{x}_{35} - \boldsymbol{x}_{12}||$ and from the second $||\boldsymbol{x}_{11} - \boldsymbol{x}_{30}||, ||\boldsymbol{x}_2 - \boldsymbol{x}_{11}||$ and $||\boldsymbol{x}_0 - \boldsymbol{x}_2||$. Here only 4 samples were used. But in the actual experiment, we used $N$ samples for each random walk, and we repeated 100 times. Then, if we calculate the histogram of these lengths for datasets of different levels of sparsity, we obtain results such as those of Fig. 4A. Now each histogram vector is fed to a regressor to predict a single value $\theta$. A straightforward 1D CNN provides 0.99 accuracy in predicting the correct $\theta$ in this example. Calculating the random walks takes only a few seconds because the complexity of calculating

the lengths is $\mathcal{O}(\text{N})$. The histogram calculation for $N$ of 1 million takes only 12 ms for 100 bins. Therefore, it does not delay execution. Finally, in order to understand why the lengths of these random walks become a signature of the underlying distribution, please see Fig. A30.

## A.14 ON THE STOCHASTICITY OF THETAN BERSERKER

An algorithm is said to be stochastic if it incorporates randomness in its process or decision-making. This means that the algorithm's behavior or output may or not vary between executions, even when given the same input. More importantly, stochastic algorithms use random variables or probabilistic components as part of their logic. Thetan Berseker (TB) arrives at similar conclusions given the random orderings of the data samples. In addition, TB employs randomness to avoid repeating the same ordering in Algorithm 2. This acts as an extra safety measure to ensure robustness.

## A.15 GAUSSIAN DISTRIBUTIONS OF VARYING SCALE

The clusters here were generated by sampling from multivariate normal distributions with varying scales and with a mean of 0. Each distribution contains 500 samples. The centers of the distributions are set on a $30 \times 10$ grid with the neighboring centers separated by a distance of 5 units. Random scaling factors are applied to the cluster centers to introduce variability, ensuring that each cluster exhibits slightly different spreads. The total number of points is 150,000. Here, the blue crosses indicate the ground truth centroid and the red crosses indicate the estimated centroids. When the red crosses are not visible, it means that they exactly match the ground truth (blue crosses), i.e., they are under the blue crosses.

Figures A32-A36 highlight the performance of various clustering methods in predicting the centroids in Gaussian distributions of varying sizes. The results indicate that TB, HDBSCAN, and Meanshift accurately identified the centroids, while KMeans++ and DBSCAN had minor inaccuracies, misidentifying only a few centroids or adding some extra ones.

Here, the parameters used were $\theta$ 3.6 for TB, K 300 for KMeans++, eps 0.5 and min_samples 40 for DBSCAN, min_samples 40 for HDBSCAN and bandwidth 2.6 for MeanShift.

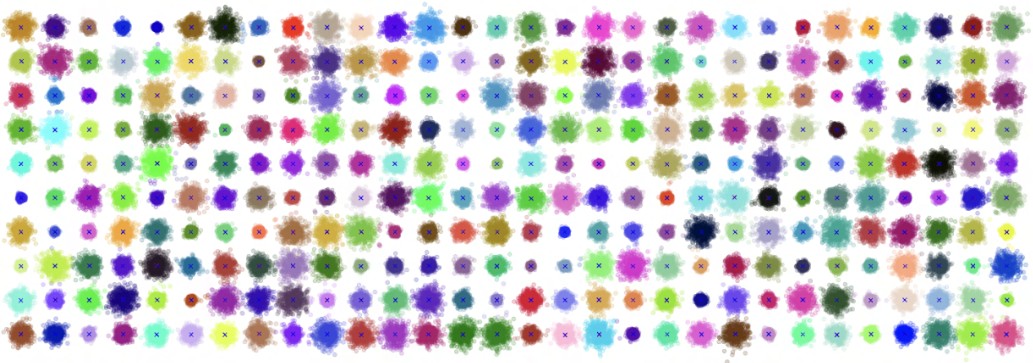

Figure A31: TB ($\theta$ 3.6)

Table A5: Comparisons between clustering algorithms on Gaussian Distribution of Varying Scale. Highlight identifies top performers.

|  | AC | | NMI | | SIL | | FMS | | ARS | | Clusters | | Duration | | Memory | |
|---|---|---|---|---|---|---|---|---|---|---|---|---|---|---|---|---|
|  | mean | std | mean | std | mean | std | mean | std | mean | std | mean | std | mean | std | mean | std |
| CURE | 0.999 | 0.0016 | 0.9963 | 0.0002 | 0.7419 | 0.0053 | 0.9927 | 0.0005 | 0.9926 | 0.0005 | 300 | 0 | 2238.3480 | 184.0060 | 174 | 0 |
| TB | 0.9993 | 0.0021 | 0.9965 | 0.0002 | 0.7435 | 0.0053 | 0.9937 | 0.0005 | 0.9936 | 0.0005 | 300.2 | 0.6325 | 0.3270 | 0.0017 | 9 | 0 |
| TBK | 1 | 0 | 0.9965 | 0.0003 | 0.7436 | 0.0053 | 0.9938 | 0.0005 | 0.9938 | 0.0005 | 300 | 0 | 0.8437 | 0.0035 | 18 | 0 |
| HDBSCAN | 1 | 0 | 0.9817 | 0.0017 | 0.7209 | 0.0072 | 0.9036 | 0.0144 | 0.9017 | 0.0151 | 300 | 0 | 130.8935 | 0.5245 | 184 | 0 |
| KMEANS++ | 0.9377 | 0.0144 | 0.9927 | 0.0012 | 0.7300 | 0.0074 | 0.9763 | 0.0051 | 0.9762 | 0.0052 | 300 | 0 | 2.3644 | 0.1082 | 20 | 0 |
| MEANSHIFT | 1 | 0 | 0.9965 | 0.0003 | 0.7436 | 0.0053 | 0.9938 | 0.0005 | 0.9938 | 0.0005 | 300 | 0 | 582.3422 | 12.8045 | 40.3 | 2.7101 |

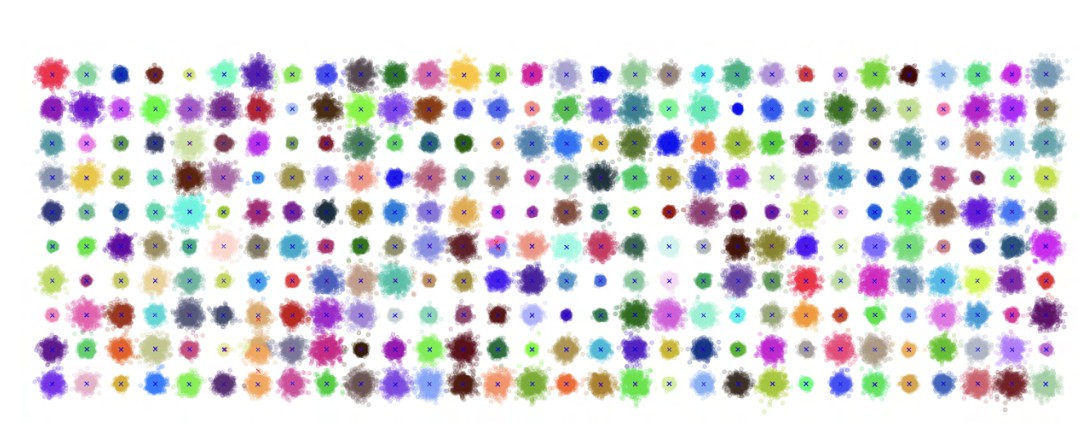

Figure A32: KMeans++ (K 300)

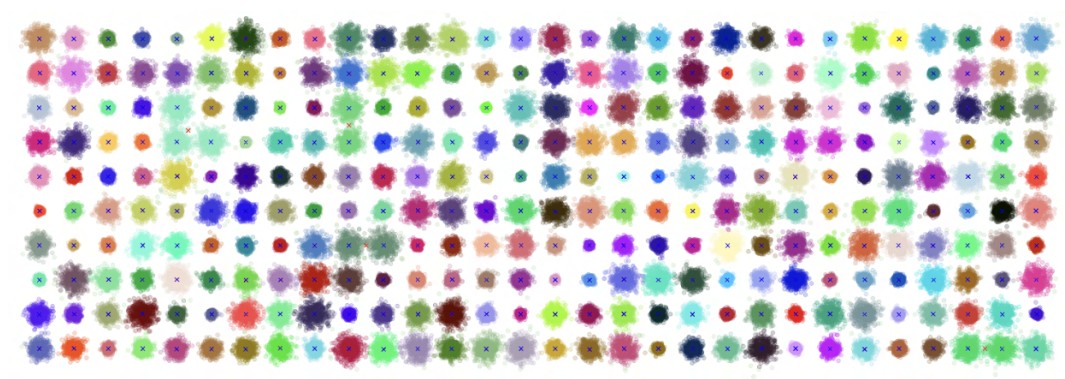

Figure A33: DBSCAN (eps 0.5, min_samples 40)

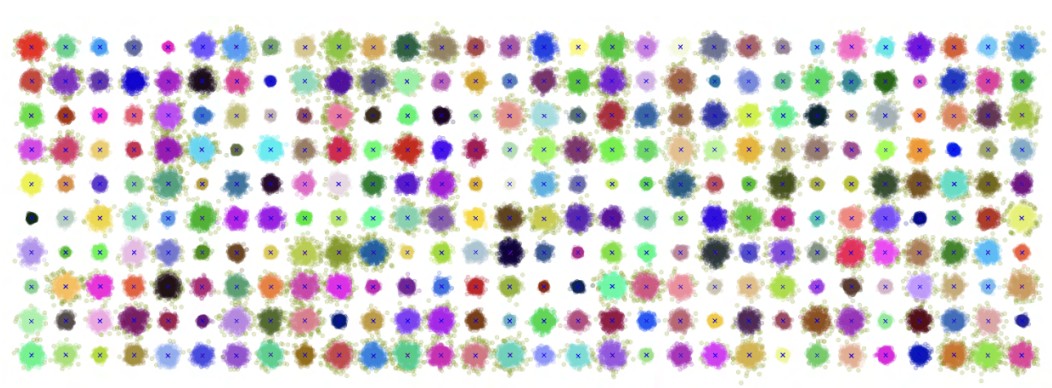

Figure A34: HDBSCAN (min_samples 40)

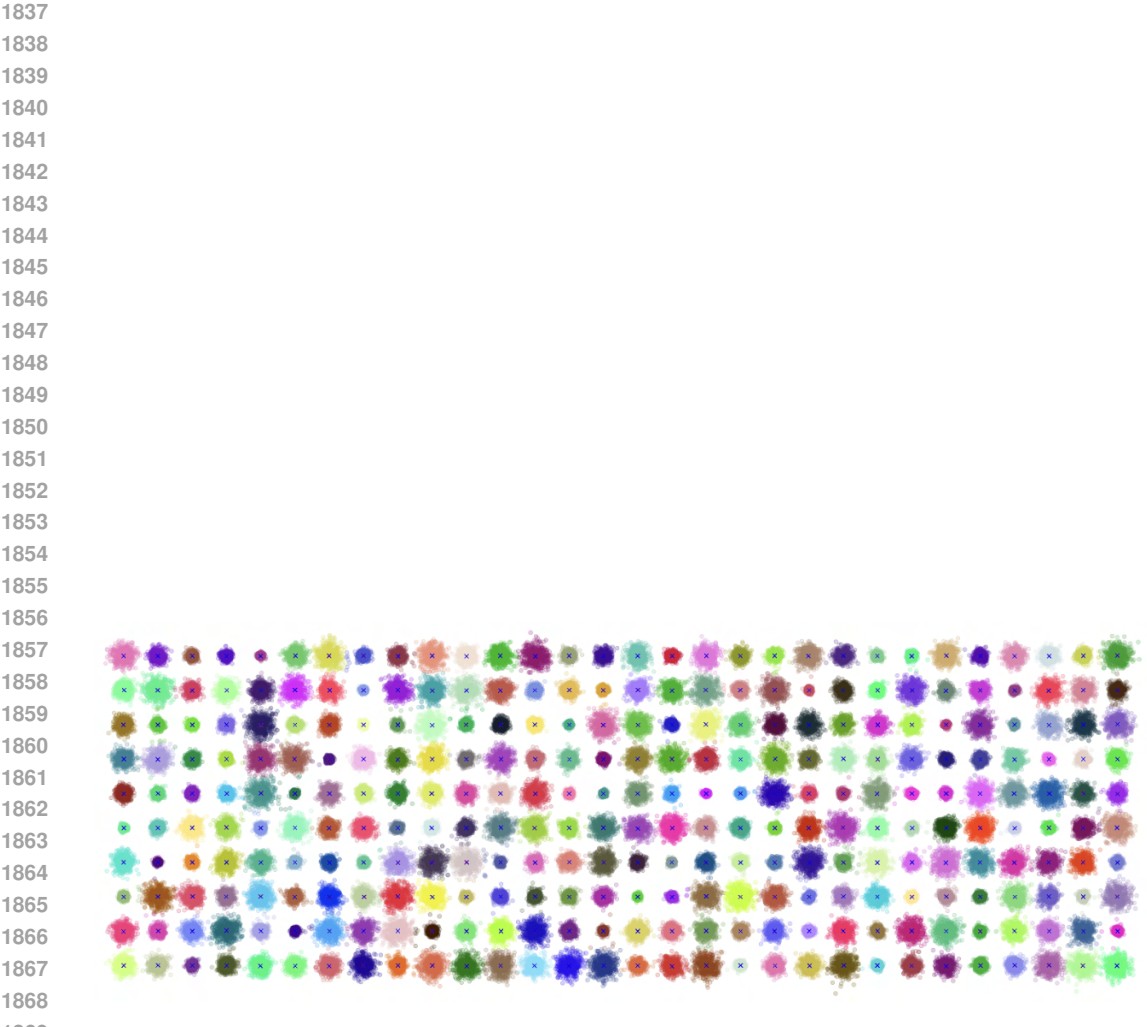

Figure A35: MeanShift (bandwidth 2.6)

## A.16 UNIFORM DISTRIBUTIONS

Here, clusters were created by sampling points uniformly within a bounded region around predefined centers. The position of each point in a cluster was determined by adding random noise to the cluster center. The noise was generated within a square region centered on each cluster, with the size of the region controlled by a scaling factor. Here, a scaling factor of 7 is used. The noise for each point was scaled by a factor alpha, which is set to 1 here, and the points were randomly distributed within the region, where the x and y coordinates of each point varied uniformly in both directions by a random value between -0.5 and 0.5. Each distribution contains 500 samples. The cluster centers are arranged on a $30 \times 10$ grid, with adjacent centers spaced 10 units apart. Therefore, the total number of ground truth clusters is 300. The total number of points in this experiment is 150,000. Here also, the blue crosses represent ground truth centroids while red crosses represent centroids obtained from the clustering method.

Figures A37 to A41 demonstrate the performance of various clustering methods in predicting the centroids of uniform distributions. The results show that TB, MeanShift, and DBSCAN accurately identified the centroids, whereas HDBSCAN and KMeans++ misidentified some of them.

The parameters used were $\theta$ 5.6 for TB, K 300 for KMeans++, eps 0.5 and min_samples 40 for DBSCAN, min_samples 40 for HDBSCAN and bandwidth 4.6 for MeanShift.

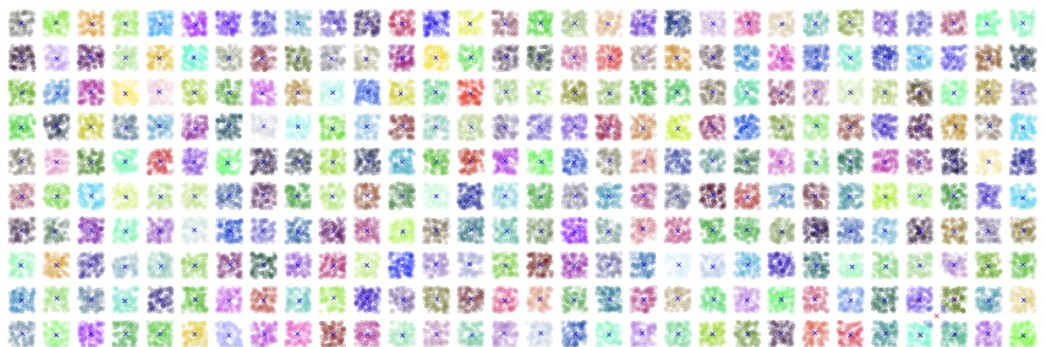

Figure A36: TB ($\theta$ 5.6)

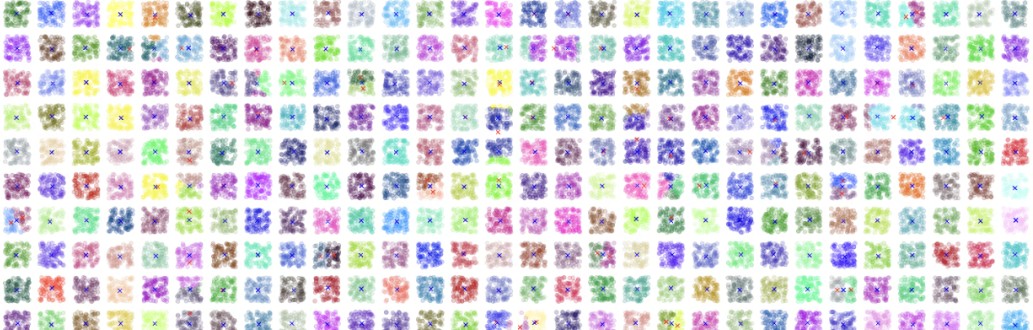

Figure A37: KMeans++ (K 300)

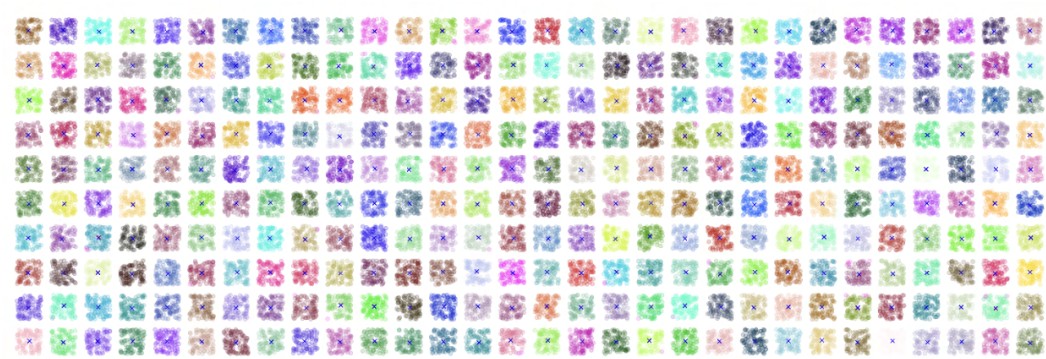

Figure A38: DBSCAN (eps 0.5, min_samples 40)

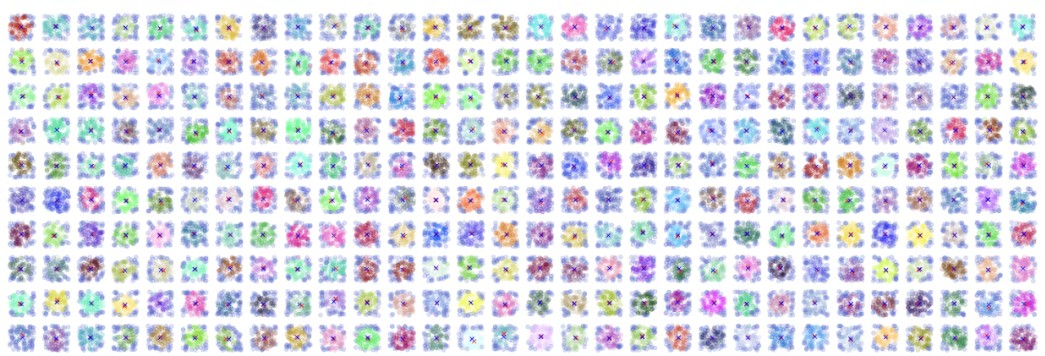

Figure A39: HDBSCAN (min_samples 40)

Table A6: Comparisons between clustering algorithms on Uniform Distribution. Highlight identifies top performers.

| | AC↑ | | NMI↑ | | SIL↑ | | FMS↑ | | ARS↑ | | Clusters | | Duration↓ | | Memory↓ | |
|---|---|---|---|---|---|---|---|---|---|---|---|---|---|---|---|---|
| | mean | std | mean | std | mean | std | mean | std | mean | std | mean | std | mean | std | mean | std |
| CURE | 0.998 | 0.004499657 | 1.0000 | 7.64591E-05 | 0.5443 | 0.0012 | 0.9999 | 0.0002 | 0.9999 | 0.0002 | 300 | 0 | 12.3378 | 0.2174 | 36 | 0 |
| TB | 0.999 | 0.00225 | 0.9999 | 0.000057 | 0.5436 | 0.0011 | 0.9998 | 0.000139 | 0.9998 | 0.00014 | 300 | 0 | 0.04172 | 0.000961 | 1 | 0 |
| TS | 0.7528 | 0.0216 | 0.9895 | 0.00107 | 0.5164 | 0.00281 | 0.9678 | 0.0035 | 0.9675 | 0.0035 | 317.5 | 1.7795 | 0.0165 | 0.0001 | 0 | 0 |
| TBK | 0.6047 | 0.0227 | 0.9730 | 0.002006185 | 0.4839 | 0.0046 | 0.8824 | 0.0095 | 0.8815 | 0.0097 | 300 | 0 | 0.2031 | 0.0077 | 3 | 0 |
| HDBSCAN | 0.1023 | 0.0211 | 0.7621 | 0.004299786 | 0.1303 | 0.0086 | 0.1029 | 0.0025 | 0.0297 | 0.0015 | 300 | 0 | 2.9633 | 0.0146 | 36 | 0 |
| KMEANS++ | 0.8353 | 0.0264 | 0.9899 | 0.001685206 | 0.5208 | 0.0030 | 0.9561 | 0.0072 | 0.9558 | 0.0072 | 300 | 0 | 0.2792 | 0.0327 | 4 | 0 |
| MEANSHIFT | 0.946 | 0.0110 | 1 | 0 | 0.5444 | 0.0012 | 1 | 0 | 1 | 0 | 300 | 0 | 49.2709 | 0.5843 | 7.9 | 0.3162 |

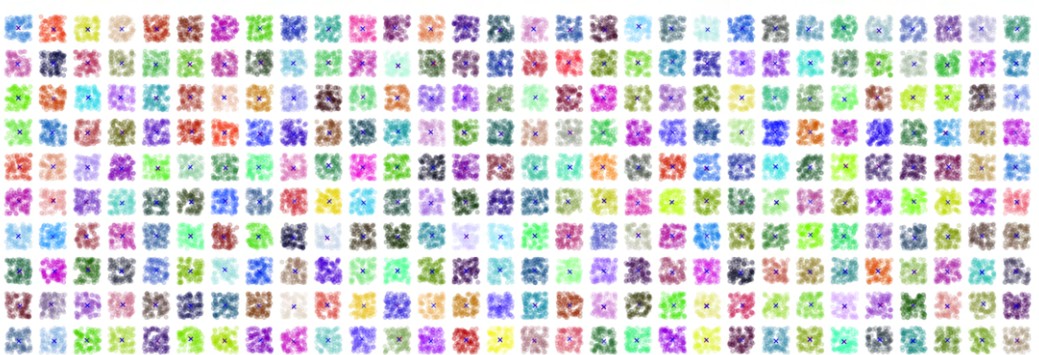

Figure A40: MeanShift (bandwidth 4.6)

### A.17 LARGE SIMULATION EXPERIMENT ON VARYING UNIFORM DISTRIBUTIONS

Similar to section A.16, the clusters were created by sampling points uniformly within a bounded region. The only difference is that the noise for each point was randomly scaled by factor alpha, which was uniformly sampled from the range of

$$0.2, 1$$

. This controls the spread of the cluster. The grid of $30 \times 10$ remained the same along the distance between the adjacent centers being 10 units apart. The total number of clusters was 300, and the number of samples per distribution was still 500, thus making the total number of points 150,000. Here also the blue crosses represent ground truth centroids while red crosses represent centroids from clustering methods.

Figures A42 to A46 illustrate the performance of various clustering methods in predicting centroids in varying uniform distribution. The results indicate that TB, DBSCAN, and HDBSCAN successfully identified the correct centroids, whereas KMeans++ and Meanshift misidentified some centroids. Similar to section A.16, the parameters used were $\theta = 5.6$ for TB, K 300 for K-Means++, eps 0.5 and min_samples 40 for DBSCAN, min_samples 40 for HDBSCAN and bandwidth 2.6 for MeanShift.

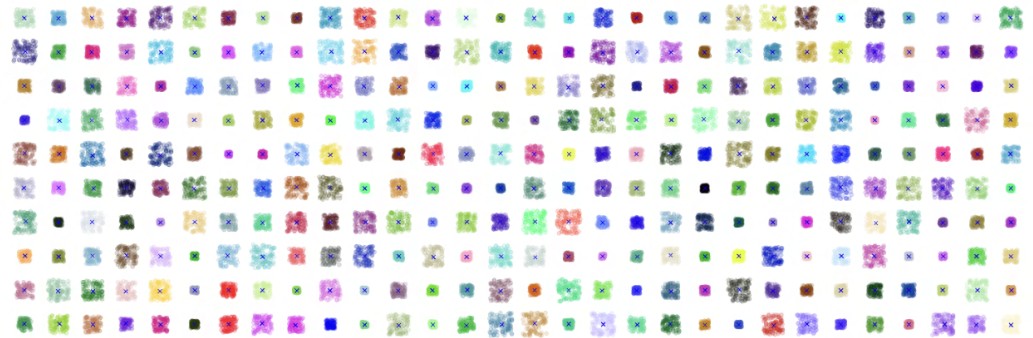

Figure A41: TB ($\theta$ 5.6)

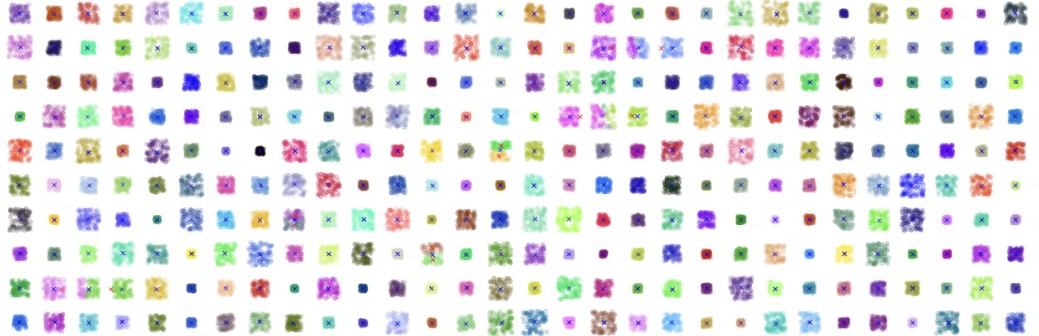

Figure A42: KMeans++ (K 300)

Table A7: Comparisons between clustering algorithms on Uniform Distribution of Varying Scale. Highlight identifies top performers.

| | AC↑ | | NMI↑ | | SIL↑ | | FMS↑ | | ARS↑ | | Clusters | | Duration↓ | | Memory↓ | |
|---|---|---|---|---|---|---|---|---|---|---|---|---|---|---|---|---|
| | mean | std | mean | std | mean | std | mean | std | mean | std | mean | std | mean | std | mean | std |
| CURE | 1 | 0 | 1 | 0 | 0.7427 | 0.0044 | 1 | 0 | 1 | 0 | 300 | 0 | 12.2249 | 0.2217 | 36 | 0 |
| TB | 1 | 0 | 1 | 0 | 0.7423 | 0.0068 | 1 | 0 | 1 | 0 | 300 | 0 | 0.0363 | 0.0006 | 1 | 0 |
| TS | 0.9756 | 0.0140 | 0.9991 | 0.0005 | 0.7376 | 0.0096 | 0.9972 | 0.0018 | 0.9972 | 0.0018 | 302.2 | 1.8135 | 0.0160 | 0.0002 | 0 | 0 |
| TBK | 0.9263 | 0.0328 | 0.9952 | 0.0023 | 0.7294 | 0.0092 | 0.9790 | 0.0097 | 0.9789 | 0.0097 | 300 | 0 | 0.0907 | 0.0043 | 3 | 0 |
| HDBSCAN | 0.9470 | 0.0088 | 0.9935 | 0.0016 | 0.7284 | 0.0064 | 0.9713 | 0.0112 | 0.9712 | 0.0113 | 299.8 | 0.421637021 | 2.9937 | 0.0187 | 36 | 0 |
| KMEANS++ | 0.9313 | 0.0188 | 0.9957 | 0.0011 | 0.7283 | 0.0050 | 0.9802 | 0.0045 | 0.9801 | 0.0045 | 300 | 0 | 0.2180 | 0.0286 | 4 | 0 |
| MEANSHIFT | 0.9973 | 0.0021 | 1 | 0 | 0.7427 | 0.0044 | 1 | 0 | 1 | 0 | 300 | 0 | 27.5586 | 0.3935 | 6.7 | 0.4830 |

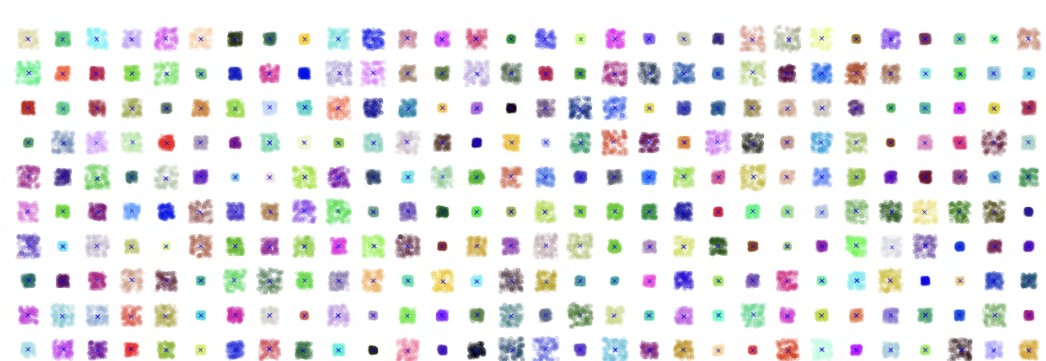

Figure A43: DBSCAN (eps 0.5, min_samples 40)

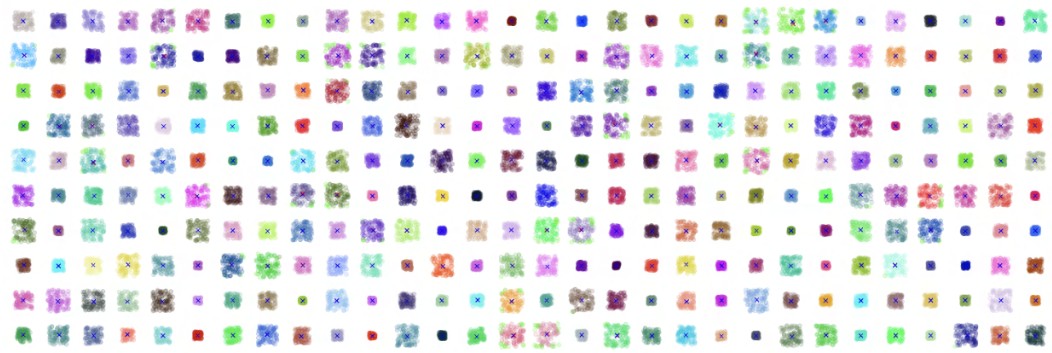

Figure A44: HDBSCAN (min_samples 40)

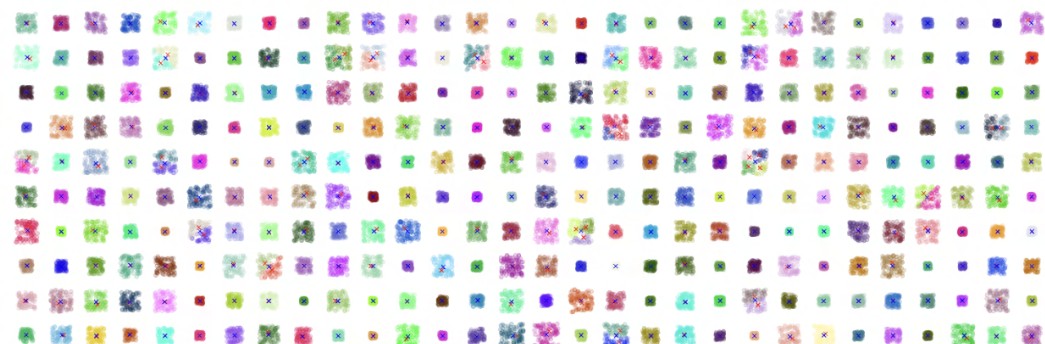

Figure A45: Meanshift (bandwidth 2.6)

## A.18 WHITE MATTER RECONSTRUCTION

Building on the discussion in 6.2, this section presents a processed slice from the T1-weighted images. The preprocessing steps and dimensions remain consistent with the earlier approach. Here, we used TB, KMeans, and SLIC.

In the A46, $A$ is the original T1 slice, $B$ is the image obtained using TB with $\theta$ 220, $C$ is the image obtained using KMeans with K 4, $D$ is the image obtained using SLIC with n_segments 10 and compactness 0.1, $E$ is TB with $\theta$ 50, $F$ is KMeans with K 310, and $G$ is SLIC with n_segments 100 and compactness 0.1.

TB does an excellent job in reconstructing the original T1 image for both values of $\theta$, outperforming both KMeans and SLIC. Interestingly, KMeans with K 310 produces results very similar to TB with $\theta$ 50. However, for a lower number of clusters, KMeans is not performing too well. It reconstructs most of the image but with some loss of information (e.g. edge loss). The number of clusters for KMeans was taken directly from TB's output. On the other hand, SLIC is not performing too well; it is losing contrast in a lower number of segments, and the loss of information is high when the number of segments is 100 (high loss of contrast and edges).

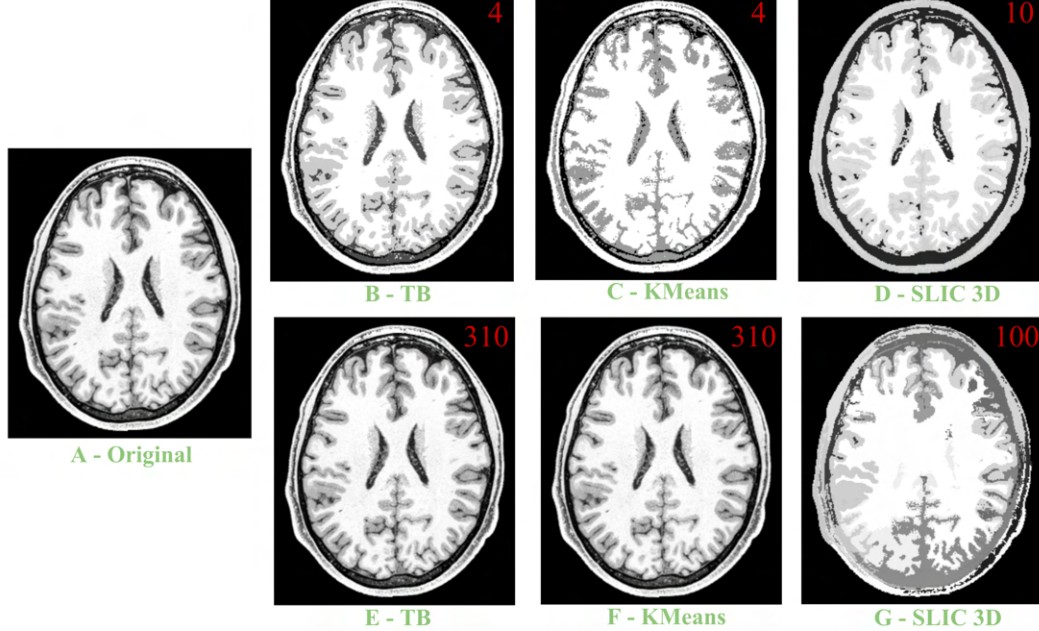

Figure A46: White Matter Reconstruction using Different Methods A) Original T1 Slice, B) TB ($\theta$ 220), C) KMeans (K 4), D) SLIC (n_segments 10), E) TB ($\theta$ 50), F) KMeans (K 310), G) SLIC (n_segments 100). The number of generated clusters is shown with red. TB is reconstructing the image more faithfully to the original image than KMeans and SLIC 3D.

## A.19 TEXT EMBEDDINGS

Here, TB was used for clustering in high-dimensional spaces, using text embeddings from the "PersonaHub FineWeb-Edu 4 Clustering 100k" dataset available on HuggingFace. The dataset comprises 100,000 samples, each represented by embeddings of dimensionality 1024. The performance of TB was benchmarked against KMeans++ and Marigold (Mortensen et al., 2023).

Tab. A8 highlights that TB requires less memory compared to both KMeans++ and Marigold when clustering the same number of clusters. In terms of execution time, TB outperforms KMeans++ but is marginally slower than Marigold. However, TB demonstrates superior clustering quality, achieving higher NMI and V-Measure scores than both KMeans++ and Marigold, while Marigold lags behind in these metrics.

For TB, the parameters used were $\theta = 0.48$, and the number of clusters derived from TB was applied to both KMeans++ and Marigold to ensure a fair comparison across methods.

Table A8: Comparisons between clustering algorithms for Text Embeddings

| Method | Clusters | Memory (MB) ↓ | Time (s) ↓ | NMI ↑ | V-Measure ↑ |
|--------|----------|---------------|------------|-------|-------------|
| TB | 5641 | **156** | 85.9667 | **0.5252** | **0.5252** |
| KMeans++ | 5641 | 166 | 89.2508 | 0.5239 | 0.5239 |
| Marigold | 5641 | 312 | **82.2857** | 0.5130 | 0.5130 |

### A.20 1D SIGNAL DENOISING

Here, we demonstrate the use of TB in denoising one-dimensional signals. For this purpose, a synthetic signal consisting of a sine wave and random Gaussian noise was used. The signal, which became the feature space, was given a sliding window of 98. The signal was then normalized by scaling the data. Fig. A47, shows the results obtained after TB was used to denoise the signal. The reconstructed sine wave is nearly identical to the clean, noise-free sine wave. The parameter that was used is $\theta$ 174.

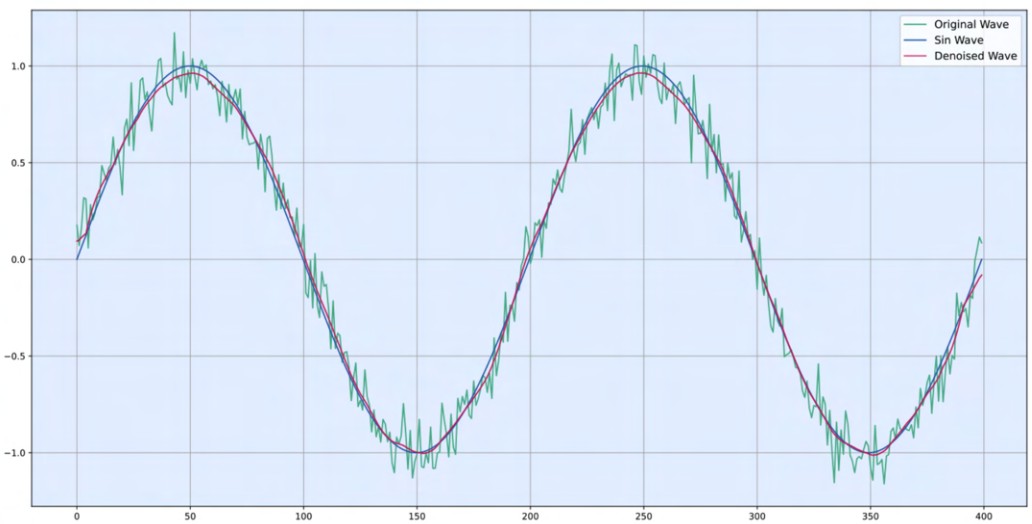

Figure A47: A sine wave denoised using TB clustering.

### A.21 GAUSSIAN DISTRIBUTIONS WITH OUTLIERS

In this experiment, TB was given a Gaussian distribution with 300 clusters and 150,000 data points. An additional 15,000 random outliers were added, bringing the total number of data points to 165,000. For each centroid, the frequency of the corresponding label was checked to see if it exceeded a certain threshold (300 in this case). These centroids are shown as red crosses on the plot, while the ground truth is indicated by blue crosses.

Figures A48 to A50 present the results when different clustering algorithms were applied to this dataset. TB performed exceptionally well on all metrics, requiring the least time and memory compared to KMeans++ and MeanShift. KMeans++ was tested with two versions: one using the correct K value (300), and the other with K set to the total number of clusters identified by TB. KMeans++, using the total number of clusters from TB, performed the worst in comparison and did not identify any correct centroids, as shown by the absence of red crosses in Fig. A50. KMeans++ with K 300 identified very few of the clusters correctly while taking half the time of TB but consuming more memory for fewer clusters.

MeanShift almost identified the correct number of centroids. However, with an acceptance criterion (AC) threshold of 0.1, it appears that the correct centroids were found in Fig. A51, but the AC

remained low for MeanShift, as most of the centroids were not within the threshold. Additionally, MeanShift required significantly more time and memory compared to the other algorithms.

The parameters used were $\theta = 3.6$ for TB, $K = 300$ and 4782 for KMeans++, and a bandwidth of 2.6 for MeanShift.

Table A9: Comparisons between clustering algorithms on Gaussian distribution with outliers. Highlight identifies top performers.

| | AC↑ | | NMI↑ | | SIL↑ | | FMS↑ | | ARS↑ | | Clusters | | Duration↓ | | Memory↓ | |
|---|---|---|---|---|---|---|---|---|---|---|---|---|---|---|---|---|
| | mean | std | mean | std | mean | std | mean | std | mean | std | mean | std | mean | std | mean | std |
| TB | 0.9957 | 0.0045 | 0.9961 | 0.0002 | 0.4967 | 0.0153 | 0.9921 | 0.0005 | 0.9920 | 0.0005 | 4781.4 | 24.0933 | 3.2588 | 0.02357 | 10 | 0 |
| KMEANS++(1) | 0.4553 | 0.0256 | 0.9935 | 0.0007 | 0.8563 | 0.0053 | 0.9497 | 0.0052 | 0.9482 | 0.0055 | 300 | 0 | 1.6798 | 0.1723 | 22 | 0 |
| KMEANS++(2) | 0.0091 | 0.0018 | 0.8668 | 0.0006 | 0.3216 | 0.0014 | 0.4279 | 0.0017 | 0.3089 | 0.0022 | 4782 | 0 | 57.6736 | 2.6415 | 30 | 0 |
| MEANSHIFT | 0.0554 | 0.0004 | 0.9293 | 0.0001 | 0.5914 | 0.0027 | 0.4933 | 0.0001 | 0.3937 | 8.55997E-05 | 5420.1 | 48.4113 | 307.8093 | 3.5689 | 40.1 | 0.8755 |

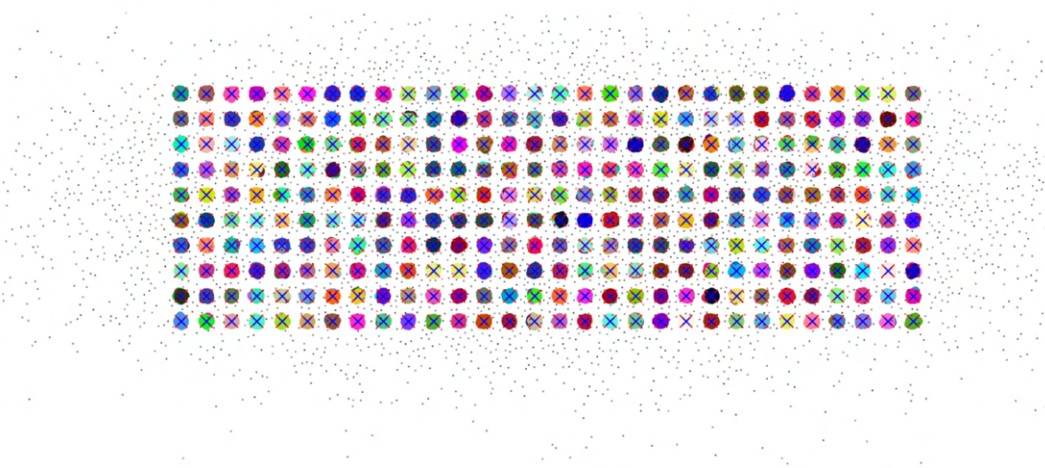

Figure A48: TB correctly finds the clusters in the presence of outliers. $\theta$ 3.6 and Duration 3.26s

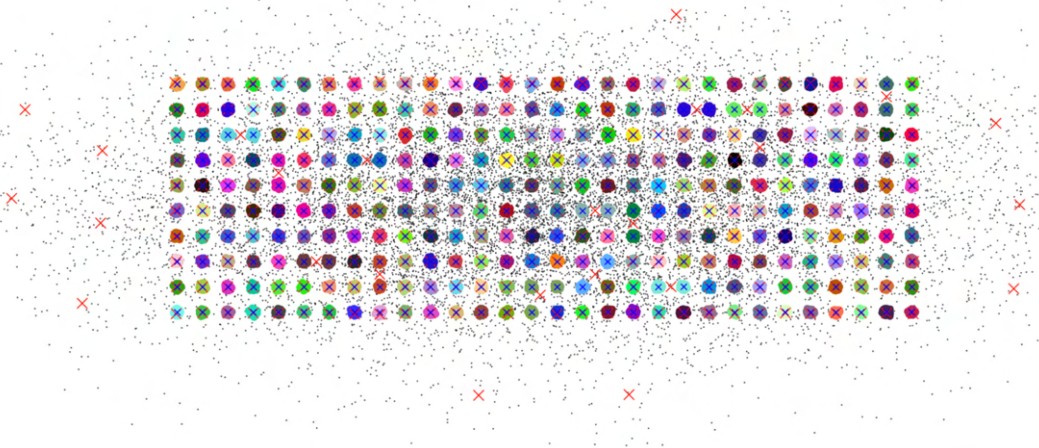

Figure A49: KMeans++ is not able to find all of the clusters in the presence of outliers. K 300 and Duration 1.68s

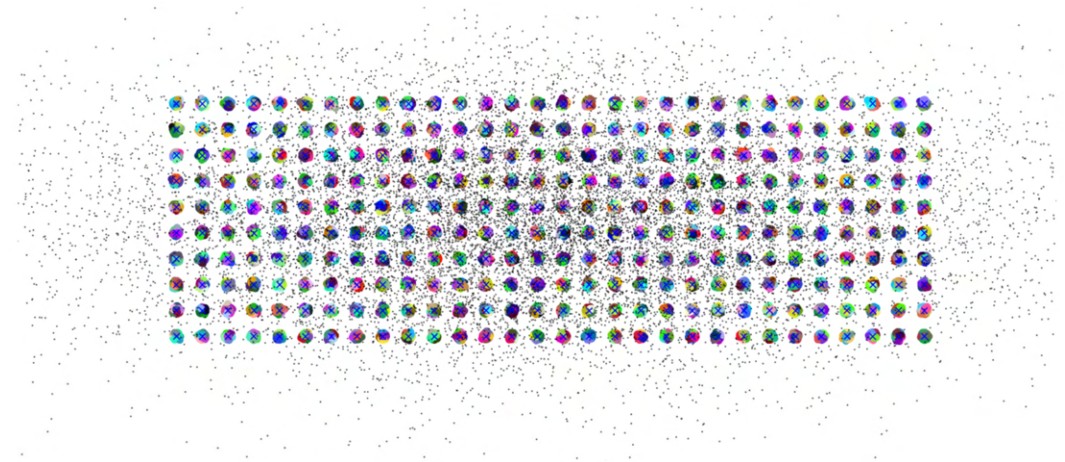

Figure A50: KMeans++ is not able to find the clusters in the presence of outliers. KMeans++ K 4782 and Duration 57.67s

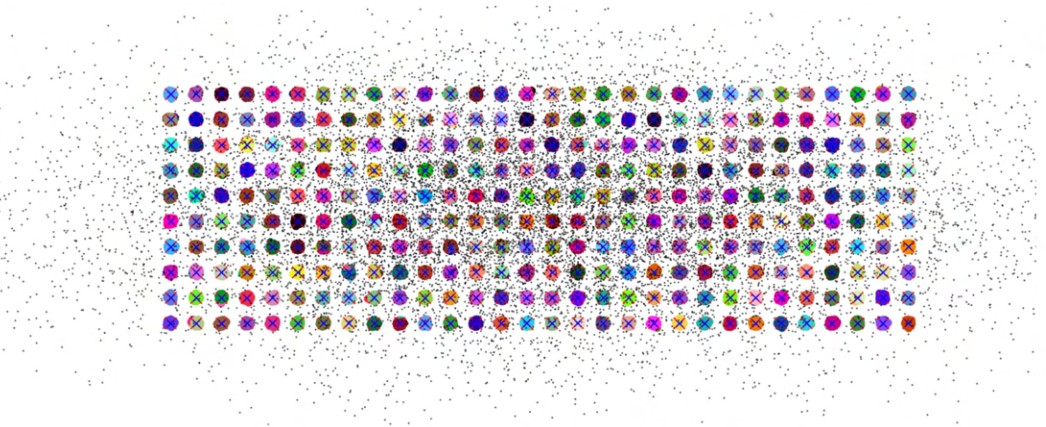

Figure A51: Meanshift is able to find most of the clusters in the presence of outliers. Meanshift bandwidth 2.6 and Duration 307.81s.

## A.22    SUMMARY OF CONTRIBUTIONS

I. Remarkably fast clustering algorithm guaranteed to be between $\mathcal{O}(n)$ and $\mathcal{O}(n^2)$.

II. Exceptionally memory efficient with best and worst case at $\mathcal{O}(n)$.

III. Outperforming in 30 experiments across datasets, dimensions and evaluation metrics.

IV. Highly interpretable consisting primarily of two compact algorithms.

V. Easy to use with only one and easy to set hyper-parameter.

VI. Used as a standalone or as a way to improve other known methods.

VII. Exceedingly robust to small samples and outliers.

