# OpenReview forum: "Thetan Berserker: Fast and Stochastic Distance-based Clustering"
_ICLR.cc/2025/Conference — ICLR 2025 Conference Withdrawn Submission_

### Official Review · Reviewer_KGjW · 2024-11-01

**Soundness:** 2
**Presentation:** 2
**Contribution:** 2
**Rating:** 5
**Confidence:** 3

**Summary:**

This paper presents a centroid-based clustering algorithm named Thetan Berserker.
This method is a parameterized by a single parameter which represents a distance threshold. This approach seems to improve the clustering metrics and runtime.

**Strengths:**

- The problem addressed in this paper is interesting and important.
- This paper uses a metaphore.

**Weaknesses:**

- The contribution is not significant enough, and the approach lacks originality.

- The paper is not self contained.

- The quality of the presentation of the paper are very poor.

**Questions:**

- The problem should be formally introduced before introducing the proposed approach, some recalls would help to make the paper self contained.

- It seems like the authors used drastically \vspaces{}, which makes the paper hard to read (for example alg1 and table 1).

- The quality of the figures should be improved, for example, the values in Figure 4 are hard to read.

- The authors stated to report the Adjusted rand score (ARS) in the first paragraph of the 5th section but this is not reported in their results.  Why ?

---

> ### Author Response · Authors · 2024-11-25
> **Thank you for your review. Version 2 of the document is now available.**
>
> > The contribution is not significant enough, and the approach lacks originality.
>
>
> The method is significant because:
>
> a) TB is a remarkably fast clustering algorithm guaranteed to be between O($n$) and O($n^2$). Please note that even k-means is a worst case superpolynomial algorithm as explained by Arthur and Vassilvitskii in their seminal 2006 paper [1].
>
> b) Exceptionally memory efficient with best and worst case at O($n$).
>
> c) Outperforming in 30 experiments across datasets, dimensions and evaluation metrics.
>
> d) Highly interpretable consisting primarily of two compact algorithms.
>
> e) Has only one hyper-parameter.
>
> f) Can be used as standalone or as a way to improve and speedup other known methods.
>
> g) Widely applicable. Examples shown in real signal, image and text datasets
>
>
> The method is original because:
>
>
> a) It is the first ever to stabilize order sensitivity at low complexity.
>
> b) It does this by only passing through the data two times. Never done before in clustering.
>
>
>
> [1] Arthur, David, and Sergei Vassilvitskii. "How slow is the k-means method?." In Proceedings of the twenty-second annual symposium on Computational geometry, pp. 144-153. 2006.
>
>
>
> > The paper is not self contained.
>
> We have worked non-stop to make this paper self-contained. We hope that you like this new version. We had to move many things to the appendix and rewrite the entire paper.
>
> > The quality of the presentation of the paper is very poor.
>
> We have improved the quality of the paper. Please look at the new version.
>
>
> > The problem should be formally introduced before introducing the proposed approach, some recalls would help to make the paper self contained.
>
> We have rewritten most of the paper. It seems easier to read now.
>
>
> > It seems like the authors used drastically \vspaces{}, which makes the paper hard to read (for example alg1 and table 1).
>
> Fixed. All vspaces removed.
>
> > The quality of the figures should be improved, for example, the values in Figure 4 are hard to read.
>
> We agree. Figure 4 has now been enlarged, separated into two parts and moved to Appendix.
>
> > The authors stated to report the Adjusted rand score (ARS) in the first paragraph of the 5th section but this is not reported in their results. Why ?
>
> Great catch. Table 1 has been updated to include Adjusted Rand Scores. ARI has also been used in later experiments.

---

> > ### Author Response · Authors · 2024-11-27
> >
> > Version 3 uploaded.

---

> > > ### Author Response · Authors · 2024-11-28
> > >
> > > Version 4 uploaded.

---

> > > > ### Author Response · Authors · 2024-11-28
> > > >
> > > > We hope that this last revision has answered all your concerns and clarified the significance and originality of the proposed approach. We are looking forward for your feedback.

---

> > > > > ### Author Response · Authors · 2024-12-02
> > > > >
> > > > > Dear Reviewer,
> > > > >
> > > > > Thank you for increasing your score for our work. We would, however, appreciate further clarification on why a score of 6 or higher might not be appropriate. We believe we have addressed all your comments and clearly demonstrated our approach's significance and originality.
> > > > >
> > > > > As you may have gathered from the paper, this is the first clustering algorithm to resolve the ordering problem with low complexity. Additionally, it demonstrates exceptional performance when handling outliers, being 100X faster than MeanShift. We have also provided a mathematical proof of its state-of-the-art runtime and memory efficiency.
> > > > >
> > > > > We have made several improvements, including correcting the excessive vertical spaces and enhancing the quality of the figures and overall presentation. Furthermore, we have added results using the Adjusted Rand Score (ARS) metric, which shows that our algorithm continues to outperform other algorithms in most cases.
> > > > >
> > > > > We would appreciate your feedback on why the score was 5 and not higher than or equal to 6.
> > > > > Thank you again for your time and valuable feedback.

---

### Official Review · Reviewer_n3tv · 2024-11-02

**Soundness:** 3
**Presentation:** 3
**Contribution:** 2
**Rating:** 5
**Confidence:** 3

**Summary:**

This paper introduces a novel centroid-based clustering paradigm that sequentially processes input samples, assigning data points to clusters based solely on a distance threshold. A common challenge with sequential clustering algorithms is the sensitivity of clustering results to the input order of the data. The proposed method, named TB (Thetan Berserker), addresses this issue by incorporating previously computed centroids with input samples to update or merge existing cluster centers. This approach enforces inter-cluster distances to exceed a predefined threshold, thereby reducing unnecessary clusters. Empirically, the method requires only a few iterations to achieve effective clustering results. The algorithm’s sequential nature ensures memory efficiency, and its limited number of iterations needed enhances time efficiency. Additionally, the proposed algorithm can be integrated with other traditional clustering methods for further refinement, introducing a new paradigm for sequential clustering.

**Strengths:**

(S1) Experiments -- The proposed method TB is simple yet effective, as demonstrated by a simulation dataset and two benchmark datasets. The experiments are conducted thoroughly, including ablation studies. These reveal key characteristics of TB, such as equidistant partitioning and rapid convergence in reducing cluster numbers.

(S2) Theoretical Depth -- Detailed theoretical proof with simple examples: The paper includes a detailed theoretical proof with simple examples and, aside from some minor confusing parts, is generally well written. The proofs of the theorems illustrate the impact of input data sequence ordering on clustering results and provide the functional assumptions for TB, specifically on selecting the distance threshold to ensure robustness against different sequence orderings of input data. The sequential processing approach for data clustering enhances the algorithm's scalability, which is beneficial.

**Weaknesses:**

(W1) The introduction and related work sections are tiny and need extension. The proposed method is a sequential algorithm with the whole dataset available beforehand. However, a clear categorization of different clustering algorithms, including TB’s position, is missing.

(W2) The paper emphasizes its minimal hyperparameter requirements, specifically needing only the distance threshold. However, KMeans also requires one major hyperparameter: the number of clusters. Setting the distance threshold for TB is as challenging as defining the number of clusters. Could you provide some tutorials on setting the appropriate distance threshold and empirical evidence on the sensitivity of the TB distance threshold to clustering performance in the appendix?

(W3) Soundness of the method: While the paper provides a theorem, it lacks a theoretical proof explaining why the method converges so quickly when merging Berserker centroids. Instead, it relies on a case study with simulation data. This leaves a gap in theoretical validation for convergence, as empirical results based on a single simulation dataset are limited.

(W4) Practical limitations of the method: The method relies on a key assumption, namely the minimum inter-cluster distance must exceed the distance threshold, and both the inter-cluster distance and the threshold must be greater than the diameters of the hyperspheres bounding the clusters. The authors do not discuss how challenging it is to find this appropriate distance threshold satisfying this assumption in detail, nor do they provide a sensitivity study of the assumption violations and potential strategies for real-world data.

(W5) Unfair comparison of TB with state-of-the-art superpixel algorithms: In Table 2, TB's boundary recall accuracy drops significantly to 0.271 when compared to SLIC, another simple yet effective algorithm for superpixel clustering. However, this result differs from the findings in the paper NSLIC: SLIC superpixels based on nonstationarity measure,' (Fig. 3a). The boundary recall accuracy for SLIC can reach at least 0.6, depending on the number of superpixels and the compactness parameter.

(W6) Formatting of the paper: The paper has a 10-page limit, and I noticed minimal line spacing between figures and paragraphs, such as between lines 124 and 125 or between lines 141 and 142.

**Questions:**

(Q1) Improvements of the description of the approach:
(a) Predicting \theta: Please clarify the method for predicting an appropriate distance threshold (page 7, line 366). The application of random walks to estimate distances across clusters is unclear. In Figure 5, different inter-class distances are generated in the simulation; intuitively, greater inter-class distances reflect more separation between points. However, how can these distributions reliably determine the distance threshold, inferring the minimum distance between points from different classes?
(b) Centroid update: How will the Berserker centroids be updated? In the pseudocode for Algorithm 1, line 136, there is a reference to “assign to closest cluster and update centroid,” but it would be helpful to specify the centroid update process. Additionally, how can TB be effectively combined with other methods, such as KMeans and DBSCAN?

(Q2) Theoretical proof of the runtime: Provide a theoretical proof of the runtime, specifically addressing why fast convergence is achieved from a stochastic perspective.

(Q3) Experiment evaluations: For methods benchmarking, we suggest hyperparameter optimization on other state-of-the-art algorithms to enable a fair comparison. If time constraints are an issue, please include the hyperparameter settings for the evaluated methods in the appendix.

(Q4) Related work: Since TB is a distance-based clustering method, it would be helpful to include a categorization of clustering methods, distinguishing between distance-based and other types, because many clustering methods depend on distance metrics

---

> ### Author Response · Authors · 2024-11-25
> **Thank you for your review. Version 2 of the document is now available.**
>
> > (W1) The introduction and related work sections are tiny and need extension. The proposed method is a sequential algorithm with the whole dataset available beforehand. However, a clear categorization of different clustering algorithms, including TB’s position, is missing.
>
> Introduction and related work are now re-written. Categories added.
>
> > (W2) The paper emphasizes its minimal hyperparameter requirements, specifically needing only the distance threshold. However, KMeans also requires one major hyperparameter: the number of clusters. Setting the distance threshold for TB is as challenging as defining the number of clusters. Could you provide some tutorials on setting the appropriate distance threshold and empirical evidence on the sensitivity of the TB distance threshold to clustering performance in the appendix?
>
> We have addressed this in the new experiments and proofs in the appendix. See section A.2, A.13 and sections on Proving runtime A.8 and memory A.9. We would also like to mention that not all datasets have a known number of clusters. Many real world datasets require but do not have the optimal number of clusters (groups). If we want to search for this with, for example, KMeans, a brute force approach is necessary.
>
> > (W3) Soundness of the method: While the paper provides a theorem, it lacks a theoretical proof explaining why the method converges so quickly when merging Berserker centroids. Instead, it relies on a case study with simulation data. This leaves a gap in theoretical validation for convergence, as empirical results based on a single simulation dataset are limited.
>
> Agreed. Runtime and memory complexity proofs added. Convergence explanation with proof added.
>
> > (W4) Practical limitations of the method: The method relies on a key assumption, namely the minimum inter-cluster distance must exceed the distance threshold, and both the inter-cluster distance and the threshold must be greater than the diameters of the hyperspheres bounding the clusters. The authors do not discuss how challenging it is to find this appropriate distance threshold satisfying this assumption in detail, nor do they provide a sensitivity study of the assumption violations and potential strategies for real-world data.
>
> We never had any issues finding good theta values. However, this is now properly addressed in A.2 and A.13.
>
> > (W5) Unfair comparison of TB with state-of-the-art superpixel algorithms: In Table 2, TB's boundary recall accuracy drops significantly to 0.271 when compared to SLIC, another simple yet effective algorithm for superpixel clustering. However, this result differs from the findings in the paper NSLIC: SLIC superpixels based on nonstationarity measure,' (Fig. 3a). The boundary recall accuracy for SLIC can reach at least 0.6, depending on the number of superpixels and the compactness parameter.
>
> Great point. The plots have been updated and expanded. See Fig. A24-A25 and section A.7. In summary, TB has higher scores (with an exception of Normalized Undersegmentation Error) compared to Meanshift, Meanshift++ and SLIC with different compactness parameters. The trend is consistent over a wide range of number of superpixels. Runtime of TB is faster than both Meanshift and Meanshift++, but slower than SLIC.
>
> The different reports in the metric scores compared to the NSLIC paper’s Fig. 3a is likely from how the metrics were calculated. BSD500 provides multiple segmentation maps per image, and our metrics are averaged over the segmentation maps, before calculating the average of the whole dataset. It is possible the authors of NSLIC chose one of the maps, resulting in a higher metric score overall.
>
> > (W6) Formatting of the paper: The paper has a 10-page limit, and I noticed minimal line spacing between figures and paragraphs, such as between lines 124 and 125 or between lines 141 and 142.
>
> Fixed.
>
>
> Continuing below.

---

> > ### Author Response · Authors · 2024-11-25
> >
> > > Questions:
> > > (Q1) Improvements of the description of the approach:
> >
> > > (a) Predicting \theta: Please clarify the method for predicting an appropriate distance threshold (page 7, line 366). The application of random walks to estimate distances across clusters is unclear. In Figure 5, different inter-class distances are generated in the simulation; intuitively, greater inter-class distances reflect more separation between points. However, how can these distributions reliably determine the distance threshold, inferring the minimum distance between points from different classes?
> >
> > Section explaining the \theta prediction with random walks added (see A.13). Although there is no perfect method for finding clustering hyperparameters, this method is very efficient. Alternatively, the elbow method can be used (see A.2). Measuring maximization of inter-class distances or minimization of intra-class distances is another approach that can be used. Both methods are well established. The random walks idea is new.
> >
> > > (b) Centroid update: How will the Berserker centroids be updated? In the pseudocode for Algorithm 1, line 136, there is a reference to “assign to closest cluster and update centroid,” but it would be helpful to specify the centroid update process. Additionally, how can TB be effectively combined with other methods, such as KMeans and DBSCAN?
> >
> > Centroid update section added (see A.11). TBK is a way to improve KMeans. You start with TB pick the K centroids of the K largest clusters and initialize KMeans. KMeans should converge very fast after that. In our comparisons we show TB outperforming KMeans++ as initialization. Similarly TBSCAN is a way to improve DBSCAN. We first break the data in small clusters using TB and then join them together using a DBSCAN approach. DBSCAN has a major issue with memory requirements but if we reduce the number of points (by using TB’s centroids) we can make the algorithm very fast and memory efficient. In future versions, we plan to use both the centroids and the number of clusters in each cluster as information to DBSCAN. It will require a careful update of DBSCAN’s implementation. But it seems like an exciting direction.
> >
> > > (Q2) Theoretical proof of the runtime: Provide a theoretical proof of the runtime, specifically addressing why fast convergence is achieved from a stochastic perspective.
> >
> > Done. Runtime and memory proved. Please see sections A.8 and A.9.
> >
> > > (Q3) Experiment evaluations: For methods benchmarking, we suggest hyperparameter optimization on other state-of-the-art algorithms to enable a fair comparison. If time constraints are an issue, please include the hyperparameter settings for the evaluated methods in the appendix.
> >
> > We report the hyper-parameters that generated the highest scores. Hyperparameters are available in the appendix when not available in the main text.
> >
> > > (Q4) Related work: Since TB is a distance-based clustering method, it would be helpful to include a categorization of clustering methods, distinguishing between distance-based and other types, because many clustering methods depend on distance metrics.
> >
> > Categories added in intro sections.

---

> > > ### Author Response · Authors · 2024-11-27
> > >
> > > Version 3 uploaded. Please also note that Fig. A24 and A25 are now A25 and A26.

---

> > > > ### Author Response · Authors · 2024-11-28
> > > >
> > > > Version 4 uploaded.

---

> > > > > ### Author Response · Authors · 2024-11-28
> > > > >
> > > > > We have carefully revised the manuscript according to your suggestions.
> > > > > Thank you again for your constructive feedback.

---

> ### Comment · Reviewer_n3tv · 2024-11-28
> **Acknowledgement of rebuttal**
>
> Thank you for your detailed response to our questions and comments regarding the paper's weaknesses. Let me appreciate that your revisions have improved the paper’s clarity and strengthened the soundness of your approach. The limited novelty and contribution, however, are not resolved. Regarding the experimental results, as shown in Table 2 of the paper, a more thorough comparison with other state-of-the-art methods reveals that your approach does not significantly outperform the competition. Specifically, your approach performs well on three out of six performance metrics on the BSDS500 dataset and only one out of six metrics on the NYUV2 dataset. Overall, we are maintaining our original rating. Our best wishes for a future resubmission!

---

> > ### Author Response · Authors · 2024-11-29
> >
> > Dear reviewer, the paper is proposing a clustering method. Not a superpixel method. Image segmentation is just one of our example applications. That TB can provide solutions that are similar to superpixels without being a superpixel algorithm is a great outcome but not the focus.
> >
> > In addition, please see the analysis plots of Figure A26. As you can see in this figure TB shows an overwhelmingly positive potential for use in segmentation tasks. See that TB has the highest Boundary Recall (best), lowest UEB (best), highest Explained Variation (best) and lowest Compactness (best). In addition, under-segmentation error and time are the lowest of the clustering methods. Even though these are significant results, this paper does not primarily focus on superpixels or segmentation. Instead, it introduces a novel clustering method distinguished by its consistently low runtime and memory usage across a broad range of experiments.
> >
> > Please also see Figure A46. Notice how TB reconstructs the image more closely than KMeans using only 4 clusters. These results are original and important.
> >
> > In summary, this is the first paper on Thetan Berserker clustering, a new, fast, accurate  and light method across many clustering simulations and benchmarks consisting of two easy to understand algorithms. Therefore, the segmentation results are important but secondary to the main focus of the paper and are shown as application examples with real data.
> >
> > We respectfully request that you reevaluate your assessment, identifying the algorithm as a clustering method rather than a superpixel method, and revise your score accordingly.

---

> > > ### Comment · Reviewer_n3tv · 2024-12-02
> > >
> > > If Table 2 cannot be considered an evaluation of the efficiency of the methods, we are missing benchmarking experiments on real-world datasets that compare your method with other state-of-the-art approaches in terms of clustering performance and runtime. Theoretically, you could have conducted clustering experiments on publicly available image datasets that are intrinsically high-dimensional real data. However, this is absent. As a result, we still do not have sufficient material to justify upgrading our evaluation of your paper.

---

> > > > ### Author Response · Authors · 2024-12-03
> > > >
> > > > > If Table 2 cannot be considered an evaluation of the efficiency of the methods, we are missing benchmarking experiments on real-world datasets that compare your method with other state-of-the-art approaches in terms of clustering performance and runtime. Theoretically, you could have conducted clustering experiments on publicly available image datasets that are intrinsically high-dimensional real data. However, this is absent. As a result, we still do not have sufficient material to justify upgrading our evaluation of your paper.
> > > >
> > > >
> > > > Dearest reviewer. We have provided comparisons for 1,000 real human brain 3D images. Please see section A.18. In the appendix we have also added real clustering benchmarks from the popular online benchmarking websites.  We have currently processed datasets up to 1,024 dimensions for text embeddings (See section A.19). Therefore, your concerns have been addressed.
> > > >
> > > > In addition, we write in section 6.1  “Despite the fact that our method is not tailored for superpixel purposes, it creates object boundary-compliant superpixels compared to other methods”. We also showed results in section A.7 comparing BSD500 and NYUV2 against clustering and superpixel methods.
> > > >
> > > > Therefore, we performed well in 2D segmentation and excellently in 3D. Both datasets were massive.
> > > >
> > > > Nonetheless, this paper should be considered an introductory theoretical clustering paper, with the experiments as potential applications. We reviewed all highly cited introductory papers of new clustering methods, and none has more experiments or evaluations than this paper.

---

### Official Review · Reviewer_yhHw · 2024-11-02

**Soundness:** 3
**Presentation:** 3
**Contribution:** 3
**Rating:** 8
**Confidence:** 4

**Summary:**

The paper proposes a major algorithm called Thetan Bereserker (TB) with 4 supporting minor algorithms (TS, TSR, TBK, TBSCAN). TB needs one parameter (distance) to perform, tries to address the order sensitivity problem and to estimate the number of cluster, while keeping the runtime low. Moreover, the method is tested in a rigorous way.

**Strengths:**

Overall, it is a strong paper and here are few points that I found interesting and strong about this paper:
1- The runtime is consistently low across the experiments
2- There is a reasonable mathematical framework to support the algorithm
3- It need only one parameter
4- It has a good coverage in the results section and ablation studies

**Weaknesses:**

There are a few aspects that the paper can be improved:
1- The organization of the paper makes it hard to read. The figure placements forces going back and forth between pages. They are either placed too early or too late. This makes them to seem out of context and irrelevant despite them being useful and informative.
2- The details such as What is the data? What is the sub-sampling strategy? How many runs were there?,... for the study on the robustness of the algorithm is a bit vague.
3- The paper lacks embedding based clustering analyses. That is, it is not obvious which combination of the algorithms would yield the best clusters on text clustering.
4- I could not find noise and outlier sensitivity discussion on the paper.

And, here is a suggestion rather than pointing a weakness:
This is more of a suggestion, but I think a brief section of limitation where the authors compile the points they suggested in the paper is really beneficial.

**Questions:**

I see that there are TBK and TBSCAN inspired by KMEANS and DBSCAN, was there any research tracks to explore spectral clustering methods as well?

---

> ### Author Response · Authors · 2024-11-25
> **Thank you for your review. Version 2 of the document is now available.**
>
> > Weaknesses:
> > There are a few aspects that the paper can be improved:
>
> > 1- The organization of the paper makes it hard to read. The figure placements forces going back and forth between pages. They are either placed too early or too late. This makes them to seem out of context and irrelevant despite them being useful and informative.
>
> Thank you. The figures are now placed in the correct locations.
>
>
> > 2- The details such as What is the data? What is the sub-sampling strategy? How many runs were there?,... for the study on the robustness of the algorithm is a bit vague.
>
> We have added a lot more information in the Appendix. Simulations of normal & uniform distributions are 10 runs.
>
> > 3- The paper lacks embedding based clustering analyses. That is, it is not obvious which combination of the algorithms would yield the best clusters on text clustering.
>
> Text embedding experiment added.
>
> > 4- I could not find noise and outlier sensitivity discussion on the paper.
>
> Noise and outlier experiments added.
>
> > And, here is a suggestion rather than pointing a weakness: This is more of a suggestion, but I think a brief section of limitation where the authors compile the points they suggested in the paper is really beneficial.
>
> Overall, the method has been a breeze to work with. Nonetheless, the method was examined only using L2 and it will suffer from the curse of dimensionality like most clustering methods. This is something that we would love to investigate more in the future.
>
>
> Questions:
> > I see that there are TBK and TBSCAN inspired by KMEANS and DBSCAN, was there any research tracks to explore spectral clustering methods as well?
>
> Spectral clustering (SC) is known to struggle with large datasets with many samples. Therefore, if it can read a simplified input by TB (i.e. less data but good approximations of the underlying distributions) it will of course be faster and less memory hungry. More experiments are required to know exactly how much SC can benefit from this.

---

> > ### Comment · Reviewer_yhHw · 2024-11-25
> >
> > Thank you for your thoughtful revision. While I believe these changes have strengthened the paper in clarity and technical content, I want to maintain my original rating as it already reflected my positive view toward the paper's fundamental contributions.

---

> > > ### Author Response · Authors · 2024-11-25
> > >
> > > Thank you for your careful review.

---

### Official Review · Reviewer_x5tC · 2024-11-04

**Soundness:** 2
**Presentation:** 2
**Contribution:** 2
**Rating:** 3
**Confidence:** 3

**Summary:**

The authors introduce Thetan Berserker, a centroid- and distance-based clustering method that is based on a sequential approach, Thetan Sequential (TS). The method is tested on low-dimensional data and as a way to find superpixels in images.

**Strengths:**

S1) The approach is straightforward and easy to follow.

S2) Time and memory efficient regarding the experiments performed on low-dimensional data.

S3) The proposed method depends on one hyperparameter only.

**Weaknesses:**

W1) Experimental evaluation is not sufficient

a) The main claim of the paper highlights speed and efficiency. However, the experiments are performed only on low-dimensional data. What does the performance look like on high-dimensional data? Please also include competitors aimed at speed, e.g.,  [0,1]. Furthermore, for comparing runtimes between different methods, they should be implemented in the same language as python implementations are per se slower than such in C or cython.

b) The comparison across different clustering algorithms is performed on only one main synthetic dataset (with different settings). Please add your competitors’ performance in Table A1 (benchmark datasets) and give the properties of the evaluated datasets.

c) It would be interesting to include the baseline TS in the experiments.

d) The presented real-world use cases, superpixel and processing 3D brains, show different issues. The results of TB finding superpixels are compared to other methods finding superpixels, but not to other standard clustering methods like meanshift that performed very well in the simulated dataset. The experiments on brain scans do not seem very stable and neither baselines nor comparative methods are included here .

W2) There is an explanation regarding how theta can be determined, but I am missing further information concerning the robustness and intuition for the choice of the parameter.

W3) The self-defined metric for “Apparent Centroid distance” (AC) is used throughout the analysis. However, its computation is unclear and not given formally - do you only consider distances that are smaller than 0.1? If so, why this “magic number”?

W4) Almost all figures have a way too small font size so that they are not readable.

[0] Mortensen, K. O., Zardbani, F., Haque, M. A., Agustsson, S. Y., Mottin, D., Hofmann, P., & Karras, P. (2023). Marigold: Efficient k-Means Clustering in High Dimensions. Proceedings of the VLDB Endowment, 16(7), 1740-1748.

[1] Carreira-Perpinán, M. A. (2006, June). Fast nonparametric clustering with Gaussian blurring mean-shift. In Proceedings of the 23rd international conference on Machine learning (pp. 153-160).

**Questions:**

Q1) How do you handle noise in the evaluation (for (H)DBSCAN/ TBSCAN results)?

Q2) How were the hyperparameters determined across the study, for the competitors as well as for your method? Which values were tried out and according to which criteria did you decide for the ones used in the end?

Q3) How robust are the results regarding different choices for theta?

Q4) How are the comparative methods chosen for the individual experiments?

---

> ### Author Response · Authors · 2024-11-25
> **Thank you for your review. Version 2 of the document is now available.**
>
> > W1) Experimental evaluation is not sufficient
>
> More than 20 experiments have been added. 30 in total.
>
>
> > a) The main claim of the paper highlights speed and efficiency. However, the experiments are performed only on low-dimensional data. What does the performance look like on high-dimensional data? Please also include competitors aimed at speed, e.g., [0,1].
>
> High dimensional experiment with 1024 dimensions added. Comparison with Marigold also added. Comparison with Marigold [0] added. [1] implementation was not available and only relevant for gaussian clusters. Time complexity in [1] is also larger than kN^2 which has no chance over our linear time on average complexity.
>
> Furthermore, for comparing runtimes between different methods, they should be implemented in the same language as python implementations are per se slower than such in C or Cython.
>
>
>
>
>
>
> All the compared methods have Python interfaces but the heavy computations are written in C (directly or via Cython) or in C++. Therefore, we do not bias methods in regards to language implementations.
>
> > b) The comparison across different clustering algorithms is performed on only one main synthetic dataset (with different settings). Please add your competitors’ performance in Table A1 (benchmark datasets) and give the properties of the evaluated datasets.
>
> We have added many new datasets and tables. The benchmarks used are publicly available and well established. Please see the Appendix for new additions. Because now we have 30 experiments, we ask you to kindly clarify if you want a table summarizing all 30 experiments.
> In summary, we have experiments with 1D signals, 10x10 and 30x10 grids of uniform and gaussian distributions of the same or varying size, with or without outliers, real images, medical 3D images and text embeddings in 1024 dimensions. For the 1D signals we process windows of up to 98 dimensions. So overall we process data at feature spaces of 2, 3, 4, 5, 7, 98, and 1024 dimensions.
>
> > c) It would be interesting to include the baseline TS in the experiments.
>
> TS baselines added (see Table 1, row 11).
>
> > d) The presented real-world use cases, superpixel and processing 3D brains, show different issues. The results of TB finding superpixels are compared to other methods finding superpixels, but not to other standard clustering methods like meanshift that performed very well in the simulated dataset. The experiments on brain scans do not seem very stable and neither baselines nor comparative methods are included here .
>
> Comparisons with other methods for superpixels and 3D brains added. Results are stable.
>
> > W2) There is an explanation regarding how theta can be determined, but I am missing further information concerning the robustness and intuition for the choice of the parameter.
>
> Intuition comes from the fact that data usually have actual dimensions, for example cells often of similar size but we do not know their number. Robustness has been addressed with subsampling. Same parameter was used down to using 4% of the data. In addition, two new sections have been added on the same topic (see sections A.2, A.13).
>
> > W3) The self-defined metric for “Apparent Centroid distance” (AC) is used throughout the analysis. However, its computation is unclear and not given formally - do you only consider distances that are smaller than 0.1? If so, why this “magic number”?
>
> Clarification added. 0.1 is sensible because the distributions have a diameter of 4 or larger. Therefore 0.1 means that the centroid is predicted very well. See section A.1.
>
> > W4) Almost all figures have a way too small font size so that they are not readable.
>
> This has been corrected by moving figures at larger scale to the Appendix.

---

> > ### Author Response · Authors · 2024-11-27
> >
> > Version 3 uploaded.

---

> > > ### Author Response · Authors · 2024-11-28
> > >
> > > Version 4 uploaded.

---

> > > > ### Author Response · Authors · 2024-11-28
> > > >
> > > > We have carefully revised the manuscript, and we hope the changes made adequately address all of your comments and suggestions. We believe these revisions have enhanced the clarity and quality of our paper, and we hope it meets your expectations for publication.

---

> > > > > ### Author Response · Authors · 2024-12-02
> > > > >
> > > > > Dear reviewer. Thank you for your patience. This is the follow up of your questions.
> > > > >
> > > > > > Q1) How do you handle noise in the evaluation (for (H)DBSCAN/ TBSCAN results)?
> > > > >
> > > > > TB is excellent in handling outliers as a separate label and hence even when TB is applied to a dataset with outliers, it still finds the correct clusters. Please see Appendix section A.21. With the case of TBSCAN, TB is used to initially get the centroids that DBSCAN uses thus reducing the amount of total time taken for getting the clusters. This is the essence of TBSCAN.  Since, the outliers are already put up as a separate label by TB, the followup of DBSCAN is made easy and hence noise can still be handled easily and the correct clusters are still there. Note that DBSCAN otherwise fails with large numbers of points due to memory limitations.
> > > > >
> > > > > > Q2) How were the hyperparameters determined across the study, for the competitors as well as for your method? Which values were tried out and according to which criteria did you decide for the ones used in the end?
> > > > >
> > > > > The hyperparameters are determined by first trying to maximize the evaluation metrics of Table 1. Or similar metrics as given in the linear/nonlinear benchmarks. We spent a lot of time trying to find the optimal parameters for each method, as we wanted to be as fair as possible.
> > > > >
> > > > >
> > > > > > Q3) How robust are the results regarding different choices for theta?
> > > > >
> > > > > We have addressed this question in section A.2. You can see that there are specific plateaus that make finding theta easy. In all 30 experiments, we never faced a problem finding good theta values.
> > > > >
> > > > >
> > > > >
> > > > > > Q4) How are the comparative methods chosen for the individual experiments?
> > > > >
> > > > >
> > > > > We chose methods that are widely used, well cited, and relevant for linearly or nonlinearly separable problems. We chose implementations that have underlying C or C++ implementations from popular packages such as scikit-learn or pyclustering so that their runtimes are comparable.
> > > > >
> > > > >
> > > > > In summary, we introduced an original algorithm that is:
> > > > >
> > > > >
> > > > > a) Remarkably fast clustering algorithm guaranteed to be between $\mathcal{O}$($n$) and $\mathcal{O}$($n^2$). With an average time that is linear time and in practice faster than kmeans.
> > > > >
> > > > > b) Exceptionally memory efficient with best and worst case at $\mathcal{O}$($n$). DBSCAN and MeanShift had issues to keep up with larger data.
> > > > >
> > > > > c) Highly interpretable, consisting primarily of two compact algorithms.
> > > > >
> > > > > d) Easy to use with only one and easy to set hyper-parameter.
> > > > >
> > > > > e) Used as a standalone or as a way to improve other known methods.
> > > > >
> > > > > f) Exceedingly robust to small samples and outliers.
> > > > >
> > > > > In addition, we provided proofs for runtime, memory and explained mathematically the reasons for fast convergence.
> > > > >
> > > > >
> > > > > TB is the first algorithm to solve order sensitivity problems in two passes of the data.
> > > > > Given the points outlined above, we believe this method presents significant value and is important for the AI community.
> > > > >
> > > > > Thank you again for your time and valuable feedback.

---

> > > > > > ### Comment · Reviewer_x5tC · 2024-12-02
> > > > > >
> > > > > > Dear authors,
> > > > > > Thank you for your elaborate answers and changes to the paper. I appreciate the effort and while they improved the paper, I will keep my scores.
> > > > > >
> > > > > > Regarding the changes in the paper, there are still some issues, open questions, and unclarities that could be solved in order to improve the paper for future submissions:
> > > > > >
> > > > > >
> > > > > > A.1: Please give a formal definition of your new measure AC, it is still not clear from the short description.
> > > > > >
> > > > > > A.3: As a DBSCAN, OPTICS, and HDBSCAN parameter, min_samples= 40 was chosen - this is rarely a good choice, see [2]. You could, instead, perform a grid search to find the best values, state how/why you chose them, or chose some value based on a heuristic.
> > > > > >
> > > > > > Figure A2 and A3:
> > > > > > Where do these Figures belong to? They are not addressed in the text of the appendix.
> > > > > > The presentation of these figures needs significant improvement.
> > > > > >
> > > > > > A.5 ECG Signal Experiment:
> > > > > > This experiment’s results are rather exemplary, where are your competitors in this experiment? Where are any evaluation measures?
> > > > > >
> > > > > > A.6 Added Benchmarks, Tables A1, A2
> > > > > > Most of the additional benchmark datasets are very easy to cluster, almost all basic competitors reach an NMI and ARI von 1.0 (as well as TB does). The runtime comparisons here have limited meaningfulness as you compare with methods that are not aimed at speed here
> > > > > >
> > > > > >
> > > > > > A.21: Experiments with outliers/noise
> > > > > > Why is the comparison here only with methods that explicitly don’t handle noise when you also have competitors like HDBSCAN that are able to handle noise?
> > > > > > Are outliers that randomly fall into the center of a cluster also labeled outliers?
> > > > > >
> > > > > >
> > > > > > Answer to Q2) - please state in such cases, which values exactly you tried out.
> > > > > >
> > > > > > Answer to Q3) That means you tried out a lot of different values for theta and chose the one for which the number of clusters was stable across several values?
> > > > > >
> > > > > > [2] Schubert, E., Sander, J., Ester, M., Kriegel, H. P., & Xu, X. (2017). DBSCAN revisited, revisited: why and how you should (still) use DBSCAN. ACM Transactions on Database Systems (TODS), 42(3), 1-21.

---

> ### Author Response · Authors · 2024-12-03
>
> > Dear authors, Thank you for your elaborate answers and changes to the paper. I appreciate the effort and while they improved the paper, I will keep my scores.
>
> > Regarding the changes in the paper, there are still some issues, open questions, and unclarities that could be solved in order to improve the paper for future submissions:
>
> > A.1: Please give a formal definition of your new measure AC, it is still not clear from the short description.
>
> The complete distance matrix between all estimated and ground truth centroids is created. We count the number of centroids close to the ground truth as being less than a pre-specified threshold (0.1). We then of course divide the count with the number of total ground truth centroids. This is what makes the metric stay in the range of 0 to 1. Note that even if the number of estimated models is higher than the ground truth centroids the metric will still stay in the correct range [0-1] because we find the closest from the estimated to the ground truth.
>
>
>
> > A.3: As a DBSCAN, OPTICS, and HDBSCAN parameter, min_samples= 40 was chosen - this is rarely a good choice, see [2]. You could, instead, perform a grid search to find the best values, state how/why you chose them, or chose some value based on a heuristic.
>
> The reported parameters are the optimal parameters found after using grid search. We used the metrics shown in Table 1 and verified the results visually.
>
>
> > Figure A2 and A3: Where do these Figures belong to? They are not addressed in the text of the appendix. The presentation of these figures needs significant improvement.
>
> This is explained in the caption. This is the same experiment as Table 1.
>
> > A.5 ECG Signal Experiment: This experiment’s results are rather exemplary, where are your competitors in this experiment? Where are any evaluation measures?
>
> This is the first paper of Thetan Berserker. Extensive comparisons for ECG data would be certainly out of scope for this work. We added this data to help the audience understand how the method works and showcase some of the abilities. This is adequate because this is the introductory paper of a novel method.
>
> > A.6 Added Benchmarks, Tables A1, A2 Most of the additional benchmark datasets are very easy to cluster, almost all basic competitors reach an NMI and ARI von 1.0 (as well as TB does). The runtime comparisons here have limited meaningfulness as you compare with methods that are not aimed at speed here.
>
> We understand your perspective, but we respectfully disagree. This is the first paper to present this method, and we believe it is important to highlight its performance against the leading techniques currently used.
>
> We would also like to clarify that the dataset we utilized is from the clustering benchmarks [3], which was specifically designed to benchmark clustering algorithms. In addition, the competing algorithms are achieving NMI and ARI scores of 1.0 because we have ensured they are using the most optimal parameters for a fair comparison.
>
> Furthermore, we want to emphasize that benchmarks are not only used to compare the results, but also to assess the efficiency of algorithms. Efficiency, in this context, includes factors such as memory usage and runtime, which are important aspects to consider. Algorithms like KMeans++ are optimized for speed, and we believe the aim here is to compare our method against others that have demonstrated strong performance. Therefore, we chose to include these well-performing algorithms for comparison. We hope this clarifies our approach and rationale.
>
> > A.21: Experiments with outliers/noise Why is the comparison here only with methods that explicitly don’t handle noise when you also have competitors like HDBSCAN that are able to handle noise? Are outliers that randomly fall into the center of a cluster also labeled outliers?
>
> HDBSCAN completely failed in this experiment due to memory limitations. Therefore, it was removed from the reports out of respect.
>
> > Answer to Q2) - please state in such cases, which values exactly you tried out.
>
> This has been already addressed. We used grid search to find the best parameters. We are more than happy to include all ranges if this work is accepted for publication.
>
> > Answer to Q3) That means you tried out a lot of different values for theta and chose the one for which the number of clusters was stable across several values?
>
> Yes!
>
>
> [2] Schubert, E., Sander, J., Ester, M., Kriegel, H. P., & Xu, X. (2017). DBSCAN revisited, revisited: why and how you should (still) use DBSCAN. ACM Transactions on Database Systems (TODS), 42(3), 1-21.
>
> Continuing below ...

---

> > ### Author Response · Authors · 2024-12-03
> >
> > In summary, we hope our responses have addressed your concerns. As the discussion period concludes on December 4, we kindly ask you to follow the guidelines provided to the authors. We share your appreciation for DBSCAN and its capabilities. However, we have substantial evidence that our approach consistently outperforms DBSCAN, particularly when working with large datasets. Additionally, the theoretical framework that TB provides is genuinely exciting for ICLR, as it allows for the capture of dense regions without the need to explicitly measure density—a significant first in the field of clustering.
> > We kindly request that you evaluate the paper according to the established rules and reviewer guidelines. Given the current length of the paper, requests for additional experiments or further clarification extend beyond the scope of the conference. We believe the work, as presented, is reproducible in its current form.
> > Thank you for your understanding.

---

> > > ### Author Response · Authors · 2024-12-04
> > >
> > > Dear reviewer, here you can find the exact code for calculating the AC metric.
> > >
> > > ```
> > > import numpy as np
> > >
> > > def ac_metric(ecentroids=None, centroids=None, thr=0.1):
> > >
> > >     D = distance_matrix(ecentroids, centroids)
> > >     closest = np.min(D, axis=1)
> > >     count = np.sum(closest < thr)
> > >     ac_score = count / len(ecentroids)
> > >     return ac_score
> > > ```
> > > ecentroids are the estimated centroids as lists of vector points.
> > >
> > > centroids are the ground truth centroids as lists of vector points.
> > >
> > > The distance matrix calculates Euclidean distances between the ecentroids and centroids. The distance matrix does not need to be square.
> > >
> > > We hope that everything is super clear now. Thank you in advance.

---

### Note · Authors · 2024-12-26

I have read and agree with the venue's withdrawal policy on behalf of myself and my co-authors.